# Chebyshev Policies and the Mountain Car Problem: Reinforcement Learning for Low-Dimensional Control Tasks

**Stefan Huber** [1]   **Hannes Unger** [1]   **Georg Schäfer** [1]   **Jakob Rehrl** [1]

## Abstract

We analytically solve the Mountain Car problem, a canonical benchmark in RL, and derive an optimal control solution, closing a gap after 36 years. This enables us to reveal two surprising insights: The optimal control is quite simple, yet modern RL agents display a large gap to optimality. Motivated by the analysis of the optimal control, we introduce Chebyshev policies as a universal (i.e. dense) class of RL policies from first principles. They can be trained as drop-in replacements of neural nets, reducing the regret by a factor of $4.18$, while requiring 277 times fewer parameters, fostering sample efficiency, explainability and real-time capability. Chebyshev policies are evaluated on further RL tasks, including a real-world non-linear motion control testbed. They consistently improve performance over neural nets with PPO, ARS and REINFORCE. Our results demonstrate how Chebyshev policies offer a compelling and lightweight alternative or addition to neural nets for low-dimensional control tasks.

## 1. Introduction

### 1.1. Motivation

Reinforcement Learning (RL) underwent a remarkable progress and became a very powerful paradigm to tackle a large variety of control and decision-making tasks (Sutton & Barto, 2018) with plenty of applications in virtual and real-world environments. At the same time, RL also faces a number of challenges, especially when applied to real-world tasks. Dulac-Arnold et al. (Dulac-Arnold et al., 2021) identified nine such challenges, including sample efficiency, explainability and interpretability, real-time capability and

[1]Josef Ressel Centre for Intelligent and Secure Industrial Automation, University of Applied Sciences, Salzburg, Austria. Correspondence to: Stefan Huber <stefan.huber@fh-salzburg.ac.at>.

*Proceedings of the $43^{rd}$ International Conference on Machine Learning*, Seoul, South Korea. PMLR 306, 2026. Copyright 2026 by the author(s).

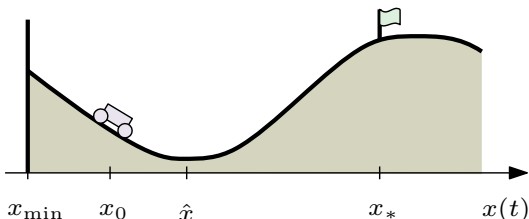

*Figure 1.* The car starts at $x_0$ and has to reach the goal at $x_*$ against gravity. There is an inelastic wall at $x_{\min}$.

training stability, see also more recent surveys by Tang et al. (Tang et al., 2025) and Gazi et al. (Gazi et al., 2026). We also lack understanding on theoretical foundations, for instance on RL training dynamics and implicit regularization (Eysenbach et al., 2023).

In this paper, we take a step back and address these fundamental challenges by revisiting a classic RL benchmark task from ground up, namely the *Mountain Car* problem (Moore, 1990), and then consequently draw conclusions from our learnings. As Figure 1 illustrates, a car shall apply minimum total propulsion to overcome gravity and reach the goal at the top. The propulsion is too small to reach the goal at once, illustrating the concept of delayed reward and exploration.

The Mountain Car problem is interesting, because the underlying system dynamics is of a typical form for physics and engineering problems, while still being simple enough to hope for an analytic treatment. Still, the optimal solution to this problem is unknown for 36 years. Consequently, the *regret* (gap to optimality) of the current state of the art (SOTA) is unknown, too, leaving a simple, yet central question wide open: *How close to optimality are current RL algorithms actually on this canonical benchmark task?*

**Contribution**   We analytically derive an optimal control solution, which allows us to evaluate the regret of the currently leading RL agents, revealing two surprises: The optimal solution is surprisingly simple and the currently top-performing RL agents display a surprisingly high regret.

Motivated by these findings, we facilitate a multi-variate generalization of Chebyshev polynomials for a novel model

of stochastic policies in RL, which we call *Chebyshev policies*. They form universal approximators in the sense that they yield a dense subset of the space of continuous policies. We evaluate them with Proximal Policy Optimization (PPO), Augmented Random Search (ARS) and REINFORCE on the Mountain Car problem and reduce the regret by a factor of 4.18, while having a factor of 277 less trainable parameters. This naturally addresses sample efficiency, explainability, interpretability and real-time capabilities, and therefore addresses five of nine key challenges of real-world RL identified by Dulac-Arnold (Dulac-Arnold et al., 2021).

We evaluate Chebyshev policies on further low-dimensional control tasks, namely the Pendulum environment of Gymnasium and the real-world helicopter-like Aero 2 testbed with non-linear dynamics. On all tested environments, Chebyshev policies clearly improve upon Multilayer Perceptron (MLP)-based policies, for both PPO and ARS, and they improve on the sim-to-real transfer and the control dynamics. Furthermore, our analysis of Mountain Car suggests two richer variants of this benchmark that mitigates some simplicities of the optimal control. All implementation, training and evaluation code is published at (Huber et al., 2026).

### 1.2. Prior Work and State of the Art

Since the original introduction of the Mountain Car problem (Moore, 1990), it evolved to different versions, e.g., concerning the landscape function, parameters defining the motion law, or the goal. To the best of our knowledge, the optimal solution to any known version of the (continuous) Mountain Car problem is unknown. In particular no analytical solution has been published so far.

We are considering the continuous Mountain Car problem as defined in (Gym-mcc). A reward of 100 is given upon reaching the goal, forming a trivial upper bound for the optimal *return* (cumulative reward), which is reduced by the propulsion applied. The Gymnasium leaderboard page (Gym-lb) considers an agent to have solved the problem when an average return of 90 for 100 consecutive trials is reached. RL Baselines3 Zoo is an RL framework for the popular RL library Stable-Baselines3 compatible to Gymnasium environments. The framework offers loading pre-trained agents, including for the continuous Mountain Car problem. Mean returns of benchmark results (Raffin et al., 2024) are in the range of 91 to 97. The current SOTA agents are trained by ARS (Mania et al., 2018), Soft Actor-Critic (SAC) (Haarnoja et al., 2018) and PPO (Schulman et al., 2017) with expected returns 96.77, 94.66, and 94.22, respectively, as reconfirmed by our own evaluation in Section 5.

These agents facilitate MLPs as approximators, typically with two hidden layers. Also models with less parameters and non-parametric models, like Gaussian Processes, have been explored for the use in RL (Grande et al., 2014).

(Rajeswaran et al., 2017) demonstrated the applicability of linear policies for continuous control tasks, but not the Mountain Car problem. (Schulman et al., 2015) propose to train a single-layer MLP on a number of random Fourier features $f(s) = \sin(\langle s, v \rangle + \varphi)$ for random vectors $v$ and phase shifts $\varphi$. However, it is unclear whether these features would be universal in the sense of forming a dense subspace of the policy space or how well they would sample it.

In a supervised learning (SL) setting, (Waclawek & Huber, 2024) demonstrated convergence capabilities of Chebyshev polynomials for piecewise $\mathcal{C}^k$-continuous approximation tasks in autodiff frameworks like PyTorch. However, we are not aware that multi-variate Chebyshev polynomials have been facilitated as parametrized models in SL or RL so far.

## 2. Analytical Solution to Mountain Car

Let us recall Figure 1: A car starts at $x_0$ in the vicinity of the minimum $\hat{x}$ and has to move to a goal $x_*$ close to the maximum, while spending a minimum amount of propulsion effort against the gravitational potential. The constants of the motion laws are such that a single monotone stroke is insufficient to reach $x_*$, but the car needs to oscillate.

The problem as coded in (Gym-mcc) can be formulated as follows: Given a sequence $(\alpha_t)$ of actions $\alpha_t \in [-1, 1]$, the trajectory $(x_t)$ of the car is governed by the system

$$x_{t+1} - x_t = v_{t+1},$$
$$v_{t+1} - v_t = a_{\max} \cdot \alpha_t - g \cdot \cos(3x_t),$$

with $a_{\max} = 0.0015$, $g = 0.0025$, and $x_t$ and $v_t$ are limited to $[x_{\min}, x_{\max}] = [-1.2, 0.6]$ and $[-v_{\max}, v_{\max}] = [-0.07, 0.07]$, respectively. The initial $x_0$ is taken from the uniform distribution $\mathcal{U}([-0.6, -0.4])$ and $v_0 = 0$. The pairs $(x_t, v_t)$ form the states and $\alpha_t \in [-1, 1]$ the actions for the agent. The objective is to maximize the *return*

$$R = -0.1 \cdot \sum_{t \geq 0} \alpha_t^2 + (100 \text{ once goal is reached})$$

over all policies $(x_t, v_t) \overset{\pi}{\mapsto} \alpha_t$, where the "goal is reached" if for some $t_* \leq t_{\max} = 999$ we have $x_{t_*} \geq x_*$ and $v_{t_*} \geq v_*$ with $(x_*, v_*) = (0.45, 0)$. Given that $R \geq 0$ is achievable, we can rephrase the problem as follows:

$$
\begin{aligned}
\min_{\pi} \quad & \ell \\
\text{s.t.} \quad & \forall t : x_t \in [x_{\min}, x_{\max}], v_t \in [-v_{\max}, v_{\max}], \quad (1) \\
& \exists t_* : x_{t_*} \geq x_* \wedge v_{t_*} \geq v_*
\end{aligned}
$$

with $\ell = \sum_{t=0}^{t_*} \alpha_t^2$, $v_0 = 0$ and $x_0 \sim \mathcal{U}([-0.6, -0.4])$. We will denote by $t_*$ the time when the goal is first reached.

We first translate the problem into a continuous setting as

follows. For $\alpha\colon [0,\infty) \to \mathbb{R}$ solve

$$\min_{\alpha} \quad \ell$$
$$\text{s.t.} \quad \exists t_* \in [0, t_{\max}]\colon\ x(t_*) \geq x_* \wedge \dot{x}(t_*) \geq v_*,$$
$$\alpha(t) \in [-1, 1], \qquad\qquad\quad (2)$$
$$x(t) \in [x_{\min}, x_{\max}], \dot{x}(t) \in [-v_{\max}, v_{\max}]$$

where

$$\ell = \int_0^{t_*} \alpha(t)^2 \, \mathrm{d}t \qquad (3)$$

and $x\colon [0,\infty) \to \mathbb{R}$ is governed by the nonlinear ODE

$$\ddot{x} = a_{\max} \cdot \alpha - g \cdot \cos(3x)$$
$$\text{with} \quad x(0) = x_0, \ \dot{x}(0) = 0. \qquad (4)$$

This optimization problem will be solved in three steps: (i) we investigate the dynamics of (4), (ii) solve the unconstrained loss minimization of (3) and (iii) reestablish the constraints given in (2).

**Step 1: Spatial Formulation of the Dynamics** In order to solve (4), a central idea is to bring it into the form $\ddot{x} = -U'(x)$. This requires the right-hand side of (4) to be determined solely by the locus $x$ and to eliminate time: we need be able to explain the action $\alpha(t)$ as a spatial *action field* $\tilde{\alpha}(x)$ instead. This form can then be approached with classic tools from calculus, including Theorem 2.1, which can be found in textbooks like (Königsberger, 2004), p. 277, and (Hand & Finch, 1998), p. 125, by the following consideration: Defining $E = \frac{1}{2}\dot{x}^2 + U(x)$, observe that

$$\frac{\mathrm{d}E}{\mathrm{d}t} = \dot{x}(\ddot{x} + U'(x)) = 0,$$

saying that $E$ is constant.[1] Then the following theorem is known for nonlinear oscillators:

**Theorem 2.1** ((Königsberger, 2004), p. 277). *Consider an interval $[e, f]$ with $U(e) = U(f) = E$ and $U'(e) \neq 0 \neq U'(f)$ and $U(\xi) < E$ for $\xi \in (e, f)$. Then a solution function $\phi$ of $\ddot{x} = -U'(x)$ periodically oscillates between $e$ and $f$ with a period*

$$T = 2 \int_e^f \frac{1}{\sqrt{2(E - U(\xi))}} \, \mathrm{d}\xi$$

*and on the interval $[0, \frac{T}{2}]$ the inverse $\phi^{-1}\colon [e, f] \to [0, \frac{T}{2}]$ is given by*

$$\phi^{-1}(x) = \int_e^x \frac{1}{\sqrt{2(E - U(\xi))}} \, \mathrm{d}\xi.$$

[1] $E$ is called "energy" and $U$ is called "potential".

Assume for a moment we place no action, i.e., $\alpha(t) = 0$ and $U'(x) = -g\cos(3x)$. Then Theorem 2.1 says that the car will oscillate forth and back periodically, $\dot{x}(e) = \dot{x}(f) = 0$, and in particular $e = x_0$ of our optimization problem.[2] Since $U(x_*) > U(x_0)$ for our given constants, the car is not reaching the goal without action: We have to apply action to lower the potential $U(x_*)$ at the goal $x_*$.

If we apply action in the direction of $\dot{x}$, i.e., $\alpha(t) \cdot \dot{x}(t) \geq 0$, then the potential $U$ is further lowered, and otherwise it is raised. We can therefore restrict our consideration to actions with $\alpha \cdot \dot{x} \geq 0$. The oscillation period of a pendulum in a potential $U$ only increases by the actions increasing the kinetic energy, which is summarized as

**Lemma 2.2.** *Denote $T$ as in Theorem 2.1 when no action is applied. For any $\alpha$ with $\alpha \cdot \dot{x} \geq 0$, the roots of $\dot{x}(t)$ are at least $\frac{T}{2}$ apart.*

Assume the car reaches the goal in finite time $t_*$ then we have finitely many roots $t_0, \ldots, t_{k-1}$ of $\dot{x}(t)$ before $t_*$ by Lemma 2.2. Note that $t_0 = 0$ and for simplicity we define $t_k = t_*$. On each interval $[t_{i-1}, t_i]$, which we call a *stroke*, we have a strictly monotone $x$. We denote $x_i = x(t_i)$, yielding $x_k = x_*$.

Let us focus on each single stroke $I_i = [t_{i-1}, t_i]$ and denote by $\phi_i$ a solution as in Theorem 2.1, which can therefore be inverted on $I_i$, i.e., $\phi_i^{-1}$ translates $x$ into $t$. This allows us to define the action field $\tilde{\alpha}_i$ over $x$ for the $i$-th stroke as $\tilde{\alpha}_i = \alpha \circ \phi_i^{-1}$. This notion of an action field now allows us to bring (4) into a purely spatial form $\ddot{x} = -U_i'(x)$, where $U_i(x) = U_{i,a}(x) + U_g(x)$ with

$$U_{i,a}(x) = -a_{\max} \cdot \int_{x_{i-1}}^x |\tilde{\alpha}_i(z)| \, \mathrm{d}z + U_{i-1,a}(x_{i-1})$$

$$\text{and} \quad U_g(x) = \frac{g}{3}\left(\sin(3x) - \sin(3x_0)\right),$$

where we define $U_{0,a}(x_0) = 0$. Note that on the $i$-th stroke the domain of $U_i$ is $[x_{i-1}, x_i]$. Also note that by choice of the integration constants, we have $U_1(x_0) = U_g(x_0) = U_{1,a}(x_0) = 0$ at the first stroke $I_1$. Also the $U_i$ of consecutive strokes meet continuously, i.e, $U_i(x_i) = U_{i+1}(x_i)$. Hence, we have $E = 0$ over all strokes. Consequently, at the last stroke $I_k$ we require $U_k(x_*) \leq -\frac{1}{2}v_*^2$ at the goal, since $E = \frac{1}{2}\dot{x}^2 + U_k(x) = 0$.

To simplify matters and to reestablish a holistic view over all strokes, we introduce a new spatial variable $\xi$ that "unrolls" the forth-and-back motion of $x$, like the car's odometer, by defining $\xi(t) = x_0 + \int_0^t |\dot{x}(\tau)| \, \mathrm{d}\tau$ and $\xi_i = \xi(t_i)$ analogously to $x_i$ and $\xi_* = \xi(t_*)$. Note that $\dot{\xi} = |\dot{x}|$ and as $\xi$ is

[2] Note that $\ddot{x} = -g\cos(3x)$ describes the mathematical pendulum; its solution is the incomplete elliptic integral of the first kind. A suitable physical interpretation of the Mountain Car problem would be a pendulum where $x$ is the angle and $\alpha(t)$ is additional torque to be applied to reach the goal.

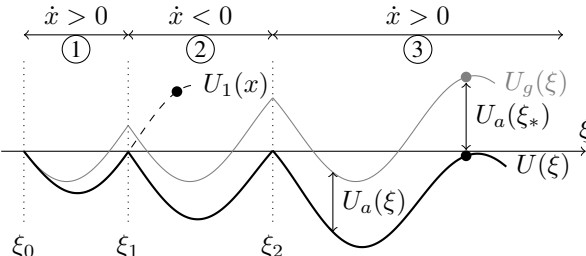

*Figure 2.* The potential $U$ and $U_g$ over $\xi$, with three strokes. The difference is $U_a$. When enough action is applied, the goal (black dot) is lowered to negative potential and hence reached at positive velocity. In dashed we extended $U_1$ beyond the 1st stroke.

invertible and we can uniquely reconstruct $x$ from $\xi$. This allows us to pull over $\tilde{\alpha}(\xi)$, $U(\xi)$, $U_g(\xi)$ and $U_a(\xi)$ to $\xi$ with a slight abuse of notation by dropping the indices $i$. In Figure 2 we visualize this construction.

To reach the goal we therefore require from the accumulation of the action $\tilde{\alpha}(\xi)$ over all strokes, which is $-U_a(\xi_*)$, to overcome the gravitational potential $U_g(x_*)$, i.e.,

$$0 \leq a_{\max} \cdot \int_{\xi_0}^{\xi_*} \tilde{\alpha}(\xi)\, \mathrm{d}\xi - U_g(x_*) - \frac{1}{2}v_*^2. \qquad (5)$$

Note that $\tilde{\alpha}(\xi) \geq 0$ for the unrolled action field. We can interpret the right-hand side of (5) as the excessive (kinetic) energy at the goal: Equality in (5) means no excessive kinetic energy is wasted.

**Step 2: Unconstrained Loss Minimization** In the next step we address the loss minimization problem in (3), but without the constraints given in (2), except that the goal has to be reached, i.e., (5) needs to be fulfilled. We first translate the loss from the time domain to the unrolled spatial domain by variable substitution $t \mapsto \xi$. Using $\dot{\xi} = |\dot{x}| = \sqrt{-2U(\xi)}$ we therefore get

$$\ell = \int_0^{t_*} \alpha(t)^2\, \mathrm{d}t = \int_{\xi_0}^{\xi_*} \tilde{\alpha}(\xi)^2 \cdot \frac{\mathrm{d}t}{\mathrm{d}\xi}\, \mathrm{d}\xi$$

$$= \int_{\xi_0}^{\xi_*} \left( \frac{\tilde{\alpha}(\xi)}{\sqrt{|\dot{x}|}} \right)^2 \mathrm{d}\xi = \int_{\xi_0}^{\xi_*} \left( \frac{\tilde{\alpha}(\xi)}{\sqrt[4]{-2U(\xi)}} \right)^2 \mathrm{d}\xi. \qquad (6)$$

One interpretation of (6) is that we pay less loss at higher velocity, i.e., we yield more kinetic work of the same action at higher velocity over a given timespan. Next we will use the following lemma for the Hilbert space $\mathcal{L}^2([\xi_0, \xi_*])$ of square-integrable functions over $[\xi_0, \xi_*]$, which is a consequence of the Cauchy-Schwarz inequality (details in Section A.1.2):

**Lemma 2.3.** *Let* $f, g \in \mathcal{L}^2([\xi_0, \xi_*])$. *Then* $\min_f \|f\|$ *s.t.* $<f, g> = 1$ *is solved by* $f = g/\|g\|^2$.

**Theorem 2.4.** *The loss $\ell$ is minimized over all goal-reaching actions by $\alpha(t) = C \cdot \dot{x}(t)$ for some constant $C$ that fulfills (5) to equality.*

*Proof.* See Section A.1.3 for all details. For a brief sketch, use equality in (5) and apply Lemma 2.3 to $f(\xi) = \tilde{\alpha}(\xi)/\sqrt[4]{-2U(\xi)}$ and $g(\xi) = a_{\max}\sqrt[4]{-2U(\xi)}/\left(\frac{1}{2}v_*^2 + U_g(\xi_*)\right)$. $\square$

Note that $\alpha(t) = C \cdot \dot{x}(t)$ has a quite simple mathematical form and, in particular, it directly admits the policy $\pi\colon (x_t, v_t) \mapsto C \cdot v_t$ with the given state and action spaces. It is actually independent of $x_t$.

However, Theorem 2.4 does not reveal the optimal $C$ but we have to search for it. When we have a small $C$ then little action is applied, so for each stroke the potential $U$ is slowly lowered and the number $k$ of strokes will be high until the goal is eventually reached, see details in Section A.4.

In Figure 3 we give a graphical interpretation of the solutions of Theorem 2.4. The minimum of $U_g$ is denoted by $\hat{x}$ and $\hat{x}_i$ is the reflection $x_i$ at $\hat{x}$, and likewise for $\hat{x}_*$. Assume we start at $x_0 < \hat{x}$ and do not apply any action. Then the trajectory oscillates cyclically between $x_0$ and $\hat{x}_0$, as indicated by the smaller ellipse-like shape. When action is applied, the trajectory bends up, causing $x_1 > \hat{x}_0$ for the first stroke. This leads to a spiraling trajectory ending at $x_*$ at velocity $v_* = 0$. If we increase $C$ slightly then we reach $x_*$ with an excessive velocity $\dot{x}(t_*) > v_*$. Also $x_{k-2}$ increases until $x_{k-2} = x_*$ and then we reach the goal in only three strokes. If we reduce $C$ then the 5-th stroke does not reach $x_*$ and we require 7 strokes to reach $x_*$, and if we reduce $C$ further we will reach $x_*$ in 7 strokes at zero velocity. If $C > 0$ becomes ever smaller then the outer ellipse will be filled with the strokes of the trajectory, with $k, t_*, \xi_*$ tending to $\infty$.

**Step 3: Reestablishing the Constraints** In the final step we reestablish the constraints given in (2). First we note that $|\dot{x}| \leq v_{\max}$ and $x \in [x_{\min}, x_{\max}]$ is enforced by the RL environment. If $\dot{x}$ or $x$ would leave their respective interval then they are limited. And when $x$ is limited, also $\dot{x}$ is set to

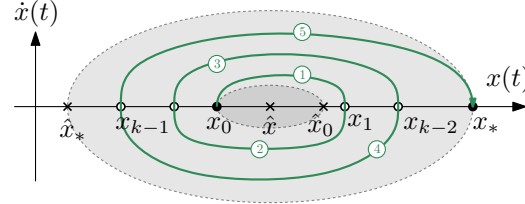

*Figure 3.* Trajectories in state space for the unconstrained optimization with $k = 5$ strokes.

zero, i.e., as an inelastic bump into a wall. We distinguish two cases:

1. The trajectory does not reach the left wall. We call this a *single-phase trajectory*.

2. The trajectory reaches the left wall. We call this a *two-phase trajectory*.

In either case we identify a finite number of candidate trajectories, depending on the value of $k$, and the optimal solution is the one with the smallest loss. In the unconstrained case we see that the smaller $C$ the smaller $\ell$, but once we hit constraints this will become unclear. We first note that we have $k \geq 2$ as $k = 1$ is impossible by (5) with the given constants. Hence, we have an upper bound $C_{\max}$ due to $k \geq 2$. Also note that there is a smallest $C_{\min}$ such that for all $C \geq C_{\min}$ we respect $t_* \leq t_{\max}$. As a consequence, there is an upper bound for $k$, which we denote by $k_{\max}$.

*Single-phase trajectories.* Consider a $C \in [C_{\min}, C_{\max}]$. Assume we reach $x_*$ at some speed $\dot{x}(t_*)$ in $k$ strokes. When we reduce $C$ then we reduce the goal velocity $\dot{x}(t_*)$, increase $t_*$, but we also reduce $\xi_* - \xi_0$ and hence $\ell$. Let us reduce $C$ until $\dot{x}(t_*) = v_*$ or $t_* = t_{\max}$, while leaving $k$ unchanged. This yields an $\ell$-optimal $C_k$ respecting $t_* \leq t_{\max}$ for all $2 \leq k \leq k_{\max}$. Figuratively speaking, we can think of starting with any $C$ large enough such that $k = 2$, and then reduce $C$ down until $C_{\min}$ while discovering all $C_k$. If $\min_t x(t) = x_{k-1}$ becomes less than $x_{\min}$ after a sufficiently large $k$ then these single-phase solutions are not feasible.

*Two-phase trajectories.* Assume $x(t)$ hits $x_{\min}$ at time $\tau$. Then the state $(x(\tau), \dot{x}(\tau))$ is reset to $(x_{\min}, 0)$ by the environment. This reset separates the future from the past, so we can discuss them independently, i.e., how to reach $x_{\min}$ and how to continue to $x_*$ in an $\ell$-optimal fashion. This establishes the *phase 1* until $\tau$ and *phase 2* after $\tau$. Observe that we have to reach the goal $x_*$ in a single stroke in phase 2, otherwise we would just reenter state $(x_{\min}, 0)$ later on.

Roughly speaking, for any given number $k$ of strokes, we search for the optimal $C_{1,k}$ for phase 1 comprising $k-1$ and the optimal $C_{2,k}$ for phase 2. In Figure 4 we have a green phase 2 trajectory that reaches $x_*$ at velocity $v_*$, which is zero. All other phase 2 trajectories have excessive velocity and reside above this trajectory. Hence, the state space is separated into the green shaded area of phase 2 trajectories and the complement, in which the phase 1 trajectories reside. This eventually allows us to formulate a single policy $\pi \colon (x, \dot{x}) \mapsto \alpha$ also for two-phase trajectories in the next section. A full discussion of two-phase trajectories and the different cases is given in Section A.2.

## 3. Optimal Policy for Mountain Car

**Single- versus Two-Phase Trajectories** A numerical search revealed that single-phase trajectories do exist. For instance, for $x_0 = -0.6$ we obtain the optimal $C = 4.891$, leading to a loss of $\ell = 5.354$, hence a of return of 99.46, and a goal velocity of zero, while not reaching the left wall. However, the best two-phase trajectories from $x_0 = -0.6$ has a slightly better return of 99.63. We experimentally found that single-phase trajectories are sub-optimal for all starting points $x_0$, limiting our discussion to two-phase policies. Furthermore, a search for the optimal two-phase trajectories showed that for each $x_0$ the optimum is achieved by having $k$ such that the left wall is reached at velocity zero and the goal at velocity zero. We determined $C_{2,k} = 4.8358$, while $C_{1,k}$ depends on $x_0$. The constraint $v_{\max}$ was never hit and can therefore be ignored for the optimal control.

**The Analytical Worst-Case Policy** The training goal of RL agents follows a Markov Decision Process (MDP) regime and hence we maximize the expected return, i.e., consider $\max_\pi \mathbb{E}_{x_0 \sim \mathcal{U}([-0.6, -0.4])}(R)$. But since the penalty of not reaching the goal is more than two magnitudes higher than the regret we have achieved, for any given $C_{1,k}$ the measure of the set of $x_0$ where the goal is not reached must be correspondingly small. Furthermore, (5) essentially tells us that the worst-case effort to reach the goal is given for $x_0$ chosen at the vicinity of $\hat{x} = -\frac{\pi}{6} \approx -0.524$. This motivates us to consider $\min_{x_0} \max_\pi R$, which we find approximately at $x_0 = \hat{x}$, and call it the *analytical worst-case policy* $\pi_{\mathrm{ana}}$.

Note that for $x_0 = \hat{x}$ we have $U_g'$ being zero and hence no action is ever applied. An arbitrarily small initial action has to be applied to "bootstrap" the motion. To overcome this hurdle, also in the presence of numerical inaccuracies, we apply a minimum action $\alpha_{\mathrm{boot}}$ in a vicinity of $\hat{x}$. This leads to the following analytical worst-case policy

$$\pi_{\mathrm{ana}}(x, v) = \mathrm{sign}(v) \cdot \max(C(x, v)|v|, \alpha_{\mathrm{boot}}(x, v)) \quad (7)$$

with $C(x, v) = 4.8358$ in the phase 2 area of the state space and $C(x, v) = 4.3346$ otherwise. We heuristically set $\alpha_{\mathrm{boot}}(x, v) = 0.1$ for $|x - \hat{x}| \leq 0.01$ and zero other-

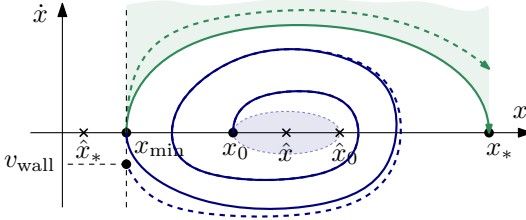

*Figure 4.* Phases 1 and 2. In the green area phase 2 trajectories can reside in. The green line gives the optimal phase 2 trajectory, and bounds the green area. The dashed trajectory has $v_{\mathrm{wall}} \neq 0$.

wise. Note that the concrete values for $\alpha_{\text{boot}}$ have small impact on $\ell$ given that only a little fraction of the state space (cf. Figure 5) is affected and the squared actions in $\ell$ are small. Hence, we refrain from further discussions in this paper. For $\pi_{\text{ana}}$ we achieve a return $R$ of 99.15 to 99.52 for $x_0 \in [-0.6, -0.4]$, see Table 1 later.

Although $\pi_{\text{ana}}$ achieves a return close to the upper bound, we can still ask for a possible gap to the original discrete formulation. For this reason we set up a discrete optimization task using the *fmincon* optimizer of Matlab R2024b and confirmed that no better solution could be found. (For the optimizer to converge, it needed to be initialized based on the continuous optimum.) See Section A.5 for details.

Based on our analysis we propose two variants of the Mountain Car problem: By moving the left wall $x_{\min}$ more to the left, at some point this would change the optimal policy from two- to one-phase trajectories. Similarly, the constraint $v_{\max}$ of 0.07 can currently be ignored, but reducing it to 0.05 would make this constraint kick in and increase the richness of this benchmark task.

**Regret of the SOTA on Mountain Car** The analytical worst-case policy $\pi_{\text{ana}}$ has a return between 99.15 and 99.52, and hence is quite close to the trivial upper bound of 100. Table 1 lists the leading agents from RL Baselines3 Zoo. The ARS agent leads with a return $R$ between 92.51 and 97.41 on different starting points $x_0$. The mean regret $r$ of ARS is therefore 2.72 and appears surprisingly high, not speaking of PPO with a regret of 5.48.

Possible explanations could be limitations in (i) the exploration phase, (ii) the RL algorithms or (iii) the (parametrized) policy space. From Section 2 we infer that a goal-reaching agent necessarily implements some form of "spiraling" trajectories in state space qua dynamics of the task, also cf. Figure 5: The state space is necessarily explored well, once the goal is reached. In RL Baselines3 Zoo, we find a large diversity of algorithms, trained with proper hyper parameter tuning and sufficient training time. On the other hand, in (Rajeswaran et al., 2017) it was demonstrated that for many well-known continuous control tasks, simple policies are beneficial to avoid overfitting models, yet Mountain Car was not investigated. Beyond the study (Rajeswaran et al., 2017), we simply see from Equation (7) and the illustration in Figure 5 that simple approximators should suffice. Hence, we ran PPO with smaller neural nets[3]. However, this led to degraded performance, see Section C.3 for details. This motivated us to look at fundamentally different approximator models from first principles.

---

[3]Default is two layers of size 64, i.e., [64, 64]. We also ran it with [32, 32], [16, 16], [64, 32] and [16].

## 4. Chebyshev Policies

### 4.1. Multi-Variate Chebyshev Polynomials

The tameness of the optimal solution, as in Equation (7), appears to be typical for low-dimensional control tasks, given the general nature of the underlying ODE. We put this principle first: A class of functions that are universal, versatile and efficient at sampling the space of continuous policies. Polynomials are versatile and an orthogonal basis spanning a dense subset of the continuous policy space provides us with universality and efficiency.

This motivated us to generalize Chebyshev polynomials to a multi-variate setting $\mathbb{R}^n \to \mathbb{R}^m$ and provide them as a parametrized model in auto-diff frameworks like PyTorch as drop-in replacements for neural nets for stochastic policies. Without loss of generality, we consider $m = 1$, since we can setup a multi-variate polynomial for each factor $\mathbb{R}$ of the co-domain $\mathbb{R}^m$.

For $(x_1, \ldots, x_n) \in \mathbb{R}^n$ the degree of the monomial $\prod_{i=0}^{n} x_i^{d_i}$ is defined in literature as $\deg f = \sum_i d_i$. A polynomial is a linear combination of monomials and its degree is the maximum degree of its monomials. For our purpose it will be convenient to introduce the notion of the *max-degree* $\deg^* f = \max_i d_i$ of a monomial and polynomial. Note that the space of multi-variate polynomials $f$ with $\deg^* f \leq d$ forms a vector space, which we denote by $P_d^n$. Also $\{\prod_{i=0}^{n} x_i^{d_i} : 0 \leq d_i \leq d\}$ forms a basis of $P_d^n$, which we call the *power basis*, and hence $\dim P_d^n = (d+1)^n$.

Let $T_k \colon [-1, 1] \to \mathbb{R} \colon x \mapsto \cos(k \cdot \arccos(x))$ denote the $k$-th Chebyshev polynomial of the first kind, having $\deg T_k = k$. Let us denote by $\mathcal{L}_w^2([-1, 1])$ the Hilbert space of square-integrable functions $[-1, 1] \to \mathbb{R}$ with the weighted inner product $\langle f, g \rangle_w = \int_{[-1,1]^n} f(x) g(x) \, w(x) \mathrm{d}x$, and the weight function $w(x) = (1 - x^2)^{-1/2}$. It is well known that the $T_0, \ldots, T_d$ form an orthonormal basis for $P_d^1$ in $\mathcal{L}_w^2([-1, 1])$, making them favorable in approximation theory, together with the property that their minima and maxima in $[-1, 1]$ have absolute value 1. Next, we generalize to $n$-variate Chebyshev polynomials and refer to (Zeller & Ehlich, 1966) for an early work on multi-variate Chebyshev polynomials and to (Dressler et al., 2024) for a more recent work on approaches for this generalization. In this paper, we use

$$T_{d_1,\ldots,d_n}(x_1, \ldots, x_n) = \prod_i T_{d_i}(x_i) \qquad (8)$$

and note that $\deg^* T_{d_1,\ldots,d_n} = \max_i d_i$. It is easy to check that the weighted orthonormality generalizes to $\langle T_{d_1,\ldots,d_n}, T_{u_1,\ldots,u_n} \rangle_w = \prod_i \delta_{d_i,u_i}$ over $[-1, 1]^n$, where $\delta_{d_i,u_i}$ denotes the Kronecker delta and the weight function is generalized to $w(x_1, \ldots, x_n) = \prod_{i=1}^{n} w(x_i)$. The $n$-variate Chebyshev polynomials also have their absolute

extreme values at 1. Note that there are $(d+1)^n$ linearly independent $n$-variate Chebyshev polynomials of max-degree at most $d$, and hence they form an orthonormal basis of $P_d^n$ again. That is, given a function $f\colon [-1,1]^n \to \mathbb{R}$ we can uniquely approximate $f$ by an element in $P_d^n$ via a linear combination

$$f \approx \sum_{i_1,\ldots,i_n=0,\ldots,0}^{d,\ldots,d} \theta_{i_1,\ldots,i_n} T_{i_1,\ldots,i_n}. \qquad (9)$$

The coefficients $\theta_{i_1,\ldots,i_n}$ could be determined by the previously explained inner product directly, but we implemented this model type in PyTorch, which allows us to facilitate them in a learning loop.

### 4.2. Chebyshev Polynomials for Stochastic Policies

We use multi-variate Chebyshev polynomials to form stochastic policies, i.e., in a given state $s$ we have a parametrized distribution $\pi_\theta(s)$ over the action space, with a parameter vector $\theta$, and we draw an action $\alpha \sim \pi(s)$. A typical implementation, like for the Mountain Car problem, is to set $\pi_\theta(s) = \mathcal{N}(\mu_\theta(s), \sigma_\theta(s))$, i.e., a normal distribution where the mean and standard deviation depend on the state. The parametrized models $\mu_\theta$ and $\sigma_\theta$ are usually implemented as a neural net of some architecture, translating $s \mapsto (\mu(s), \sigma(s))$, and $\theta$ would be the joint parameter vector of both nets.

By *Chebyshev policies* we mean that $s \mapsto (\mu(s), \sigma(s))$ is modeled by two polynomials of max-degree $d$ using multi-variate Chebyshev polynomials as basis, i.e.,

$$\mu(s) = \sum_{i_1,\ldots,i_n=0,\ldots,0}^{d,\ldots,d} \theta_{i_1,\ldots,i_n}^{(\mu)} T_{i_1,\ldots,i_n}(s) \qquad (10)$$

and likewise for $\sigma(s)$ with $\theta_{i_1,\ldots,i_n}^{(\sigma)}$. This setup makes Chebyshev policies available to all sort of policy gradient algorithms, e.g., classical REINFORCE, but also modern algorithms like Trust Region Policy Optimization (TRPO) and PPO. To this end, we implemented multi-variate Chebyshev polynomials in PyTorch in a similar manner as (Waclawek & Huber, 2024) did for the univariate case. In case of PPO, we also model the critic $v_\pi(s)$ by a Chebyshev polynomial, and hence the joint parameter vector encompasses three multi-variate polynomials. Vice versa, for ARS there is no $\sigma$ and we only have to train $\mu$.

Our practical evaluation revealed that choosing a lower max-degree $d \leq 3$ for $\sigma$ was beneficial for the training dynamics. The $\sigma$-polynomial is initialized to 1 and all other polynomials are initialized with small random coefficients in the range $\pm 10^{-3}$.

## 5. Evaluation

All experimental results were created using PyTorch version 2.4.0, Gymnasium 1.0.0, and Stable Baselines3 with RL Baselines3 Zoo versions 2.4.0. The experiments discussed in the following were conducted on the *MountainCarContinuous-v0* environment in Gymnasium.

**Do Chebyshev Policies Improve upon MLP Policies?** We evaluate agents as follows: We choose 100 evenly spaced $x_0$ from $[-0.6, -0.4]$ and record the achieved returns $R$. Note that the mean return $\overline{R}$ is an estimator for the expected return $\mathbb{E}_{x_0 \sim \mathcal{U}([-0.6,-0.4])}(R)$. Further details on the training and evaluation protocol can be found in Section C.

For neural policies we took the pretrained RL Baseline3 Zoo agents on the MountainCarContinuous-v0 problem (Raffin et al., 2024). The best performers were trained by ARS, SAC and PPO. We furthermore trained Chebyshev policies of max-degree 3 with ARS, PPO and the classical REINFORCE algorithm, which we call CH-3-ARS, CH-3-PPO and CH-3-REI. In Table 1 we report on the results and discuss them as follows. (Further details can be found in Section C.)

First of all, every Chebyshev policy trained by different algorithms significantly outperforms the neural net policies by a large margin in terms of regret. The best neural policy (ARS) achieved a regret of 2.72, while CH-3-ARS improved it to 0.65, which is a factor of 4.18. We note that for both models ARS leads to the best performer. But also with PPO, Chebyshev policies improve the regret to 1.29 from 5.48, which as again a factor of 4.24.

We actually tried to find a bad performing Chebyshev policy by implementing the classic REINFORCE algorithm. Interestingly, while REINFORCE fails in obtaining any goal-reaching neural policy (see Section C.4), it succeeds with Chebyshev policies, actually outperforming all other neural net policies by a large margin, namely improving the regret to 0.77 compared to 2.72 of ARS, which is a remarkable factor of 3.53.

*Table 1.* Performance on Mountain Car. Mean return $\overline{R}$ (regret $r$), min and max return, mean time $t_*$ to goal and $\|\pi - \pi_{\mathrm{ana}}\|_2$.

| Pol. $\pi$ | $\overline{R} \uparrow (r \downarrow)$ | min $R$ – max $R$ | $t_* \uparrow$ | $\|.\|_2 \downarrow$ |
|---|---|---|---|---|
| $\pi_{\mathrm{ana}}$ | 99.39 | 99.15 – 99.52 | 769 | |
| CH-3-ARS | 98.74 (0.65) | 98.95 – 99.11 | 471 | 0.152 |
| CH-3-REI | 98.62 (0.77) | 98.31 – 98.89 | 396 | 0.068 |
| CH-3-PPO | 98.10 (1.29) | 97.61 – 98.42 | 469 | 0.087 |
| ARS | 96.67 (2.72) | 92.51 – 97.42 | 239 | 0.211 |
| SAC | 94.61 (4.78) | 89.70 – 95.77 | 106 | 0.317 |
| PPO | 93.91 (5.48) | 90.86 – 95.23 | 298 | 0.273 |

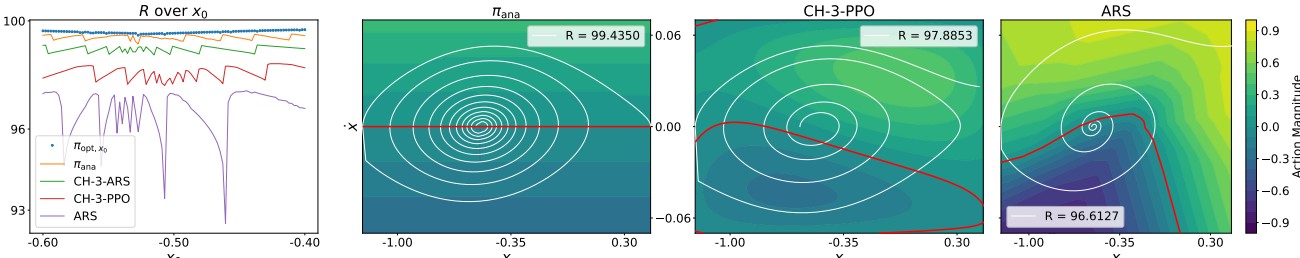

*Figure 5.* The left figure plots $R$ for each policy. The right figures plot the actions of $\pi_{\text{ana}}$, CH-3-PPO and ARS over the state space, the zero-actions in red and in white a trajectory from $x_0 = -0.55$.

**How do the Control Strategies Compare?** In knowledge of the optimal control, we can investigate deeper why and when the Chebyshev policies outperform the neural policies. In Figure 5, in the left subfigure, we plot the return of ARS (best performing neural policy) and CH-3-PPO (worst performing CH-3) over the different start positions $x_0$, and hence attain a deterministic point of view. Once we fix $x_0$, we could ask what the optimal control would be given $x_0$, which we denote by $\pi_{\text{opt},x_0}$. Interestingly, the return of ARS is highly sensitive to $x_0$, dropping down to 92.51. In contrast, Chebyshev policies give a stable performance relative to $\pi_{\text{ana}}$.

In the three right subfigures, we plot the actions over the state space and a sample trajectory. Here we can observe a key deficiency of ARS: it outputs positive action for negative $\dot{x}$ in a large area of the state space, which counteracts the dynamics of Mountain Car. ARS outputs large actions compared to $\pi_{\text{ana}}$, causing the car to reach the goal in $t_* = 239$ steps on average, lowering the return and increasing excess goal velocity, see Table 1. In contrast, CH-3-PPO is closer to $\pi_{\text{ana}}$ than ARS, which we can summarize by looking at $\|\pi - \pi_{\text{ana}}\|_2$ over the state space $[-1.2, 0.45] \times [-0.07, 0.07]$ for different polices $\pi$. See Section C.6 for more details.

**Do Chebyshev Policies Perform Well on Other Tasks?** Although Mountain Car forms a dynamical system typical for low-dimensional control tasks in engineering, we still may ask how a different dimensionality of the state space or a change of the underlying system dynamics affects the performance of Chebyshev policies. We therefore also evaluated Chebyshev policies on the *Pendulum-v1* environment in Gymnasium. Its (normalized) state space $S^1 \times [-1, 1] \hookrightarrow \mathbb{R}^3$ is 3-dimensional and formed by an angular position as a Cartesian point on the unit circle $S^1$ and the angular velocity. The action is a scalar torque applied to the pendulum. We pick $50 \times 50$ equidistant start states from $S^1 \times [-1, 1]$ and report on the average return in Table 2. The return captures a slightly different underlying objective than Mountain Car. We report on the results in Table 2 and observe that for both ARS and PPO, Chebyshev policies again clearly outperform the pretrained RL Baselines3 Zoo agents.

For the Chebyshev policies, our detailed evaluation shows that max-degree 6 worked best for ARS and max-degree 5 for PPO.

The transition from simulation to real-world environments comes at well-known RL challenges, which motivated us to evaluate Chebyshev policies on the Quanser Aero 2 helicopter testbed. Following (Schäfer et al., 2024a), we have a 3-dimensional state space, 1-dimensional action space and significant non-linear motion dynamics. In Table 2 we report returns gathered from an Aero 2 simulation and real-world operation of the same policies and the task is to follow a target pitch signal. We again see that all Chebyshev policies significantly outperform their MLP counterparts. A closer look reveals that ARS fails to follow the target signal. Furthermore, PPO resorts to a bang-bang control, leading to excessive power consumption and motor heat in our experiments.

When investigating the performance of the policies on the real system, we see that (i) all Chebyshev policies outperform all MLP policies and (ii) the Chebyshev policies better retain the simulation performance. The appendix provides more details on both environments in Section D.

**Limitations** A limitation of Chebyshev policies can be found in the combinatorial growth of the number $(d+1)^n$ of basis polynomials. Hence, for large numbers, this growth leads to high computational demands and at some point it will impair return, training stability and numerical stability. Although many industrial control tasks deal with small $n$, modern control challenges, like in humanoid robotics, deal with $n$ in the tens or hundreds. To get a preliminary impres-

*Table 2.* Average return on Mountain Car (MC), Pendulum and Aero 2 environments in simulation and real world.

|  | MC | Pendulum | Aero 2 sim | Aero 2 real |
| --- | --- | --- | --- | --- |
| CH-ARS | 98.74 | -150.8 | -125.2 | -164.2 |
| ARS | 96.67 | -218.3 | -721.8 | -718.4 |
| CH-PPO | 98.10 | -162.8 | -49.2 | -55.8 |
| PPO | 93.91 | -176.2 | -84.6 | -182.0 |

sion on computational costs, training stability and numerical limitations for $1 \leq d \leq 50$ at $n = 2$ on the Mountain Car environment, we conducted experiments in Section C.7.

Another limitation can rise from the uniform convergence of Chebyshev policies. While we prefer uniform convergence in general, it can be a weakness when the state space should not be treated equal everywhere. In contrast, the hierarchical architecture of MLPs allows to specifically focus on important regions and concentrate model capacity there. This allows MLPs to implement very different behavior at different locations, like high variety of actions at one part of the state space and low variety at other parts.

## 6. Conclusion

Reinforcement Learning is broadly applied, from particle accelerators and robotic control to large language models, yet we also face long-standing gaps in RL theory. After 36 years, we analytically solved Mountain Car to optimality. It led to surprising insights concerning the high regret of modern RL agents. This raises pressing research questions of general nature, which we underpinned with rigor evaluation on different environments.

We introduced a new class of Chebyshev policies from ground up as drop-in replacements of MLP policies from first principle: forming a universal subspace of the continuous policy space. They reduce the regret 4.18-fold, come close to optimality, and among three algorithms (ARS, REINFORCE, PPO), Chebyshev policies consistently and significantly outperformed MLP policies, while having $277\times$ less model parameters[4]. Using a $n$-variate Horner scheme, max-degree $d$ polynomials can be evaluated with $(d+1)^n$ multiplications and additions, fostering real-time performance, especially for embedded devices without neural accelerators. We also find that on the Pendulum environment and the Aero 2 real-world testbed, Chebyshev policies consistently improve upon MLP policies, improve the sim-to-real transfer and dynamical stability.

The uniform convergence of Chebyshev polynomials might also be a limitation when compared to MLPs: Especially with ReLU activation and multiple layers, a MLP can erect a policy that has a hierarchical structure that adapts to different regions of the state space. Optimal controls in resemblance to bang-bang or sliding-mode controllers are therefore better achieved by MLPs.

We anticipate numerical limitations at higher polynomial degrees and dimension, and hence our findings are currently targeted at low-dimensional control tasks. However, we suggest a future research direction on a hybrid approach

---

[4]When compared to a MLP with layer sizes 2, 64, 64, and 1, which has $2 \cdot 65 + 64 \cdot 65 + 65 = 4355$ parameters.

to combine MLPs with Chebyshev representation layers, where MLPs and Chebyshev policies form special cases. Finally, we hope that these results encourage deeper integration of analytical insight into RL, as such understanding may unlock further real-world impact.

## Acknowledgment

The financial support by the Christian Doppler Research Association, the Austrian Federal Ministry for Digital and Economic Affairs, the European Interreg Österreich-Bayern project BA0100172 AI4GREEN and the WISS project of the Federal State of Salzburg is gratefully acknowledged.

## Impact Statement

This research contributes to the development of transparent and computationally efficient RL methods. By replacing neural black-box function approximators with analytic polynomial models, it supports decision-making and control systems that are potentially easier to audit, verify, and trust. These qualities are essential for deploying RL in safety-critical domains such as robotics, energy management, and autonomous systems.

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

# A. Details of the Analytical Solution to Mountain Car

## A.1. Additional Proofs

### A.1.1. LEMMA 2.2

We investigate the oscillation period depending on the start position $x_0 = x(0)$ at rest, i.e, $\dot{x}(0) = 0$, when no action is applied. We recall that $x$ fulfills the differential equation

$$\ddot{x} = -U(x) = -g\cos(3x). \tag{11}$$

By the symmetry of $U$ we have that $U(x_0) = U(\hat{x}_0)$, where $\hat{x}_0$ denoted the reflection of $x_0$ at $\hat{x}$ and $\hat{x}$ denoted the minimum of $U$. That is, with no action the car would oscillate in the interval $[x_0, \hat{x}_0]$.

We bring the differential equation into a standard form by the substitution $\varphi = 3x + \frac{\pi}{2}$, which yields

$$\ddot{\varphi} = -3g\sin(\varphi). \tag{12}$$

Let us define $\alpha = 3x_0 + \frac{\pi}{2}$. Then $\varphi$ oscillates in the interval $[-\alpha, +\alpha]$. The period $T$ as in Theorem 2.1 is then known to yield the elliptic integral of the first kind, see p. 278 in (Königsberger, 2004). More precisely,

$$T = \frac{4}{\sqrt{3g}}K(k), \tag{13}$$

where $K$ is the complete elliptic integral of the first kind and $k = \sin(\frac{\alpha}{2})$.

### A.1.2. LEMMA 2.3

This lemma is a variant of Cauchy-Schwarz, which says that $\langle f, g \rangle \leq \|f\|\|g\|$ for elements $f, g$ of a inner-product vector space, and $\langle f, g \rangle = \|f\|\|g\|$ iff $f$ and $g$ are positive multiple of each other, i.e., the cosine between $f$ and $g$ being zero. So minimizing $\|f\|$ given $\langle f, \frac{g}{\|g\|} \rangle \leq \|f\|$ while $\langle f, g \rangle = 1$ implies $f = g/\|g\|^2$.

A geometric argument is given as follows: The set $\{f : \langle f, g \rangle = 1\}$ describes the hyperplane orthogonal to and supported by $g/\|g\|^2$. Minimizing $\|f\|$ over this hyperplane is equivalent of finding the orthogonal projection of the origin on this plane, i.e., the point $g/\|g\|^2$.

### A.1.3. THEOREM 2.4

Assume we have some $\alpha$ that reaches the goal fulfills (5) to strict inequality. Then we could simply scale $\alpha$ down by a factor, reducing $\ell$, while still reaching the goal, but with less excessive kinetic energy. Hence, we can restrict $\alpha$ to solve (5) to equality. We set

$$f(\xi) = \tilde{\alpha}(\xi)/\sqrt[4]{-2U(\xi)} \quad \text{and} \quad g(\xi) = a_{\max}\sqrt[4]{-2U(\xi)}/\left(\frac{1}{2}v_*^2 + U_g(\xi_*)\right).$$

Note that $\ell = \|f\|^2$ and minimizing $\ell$ is equivalent to minimizing $\|f\|$. Further note that by equality in (5) we have $\langle f, g \rangle = 1$. From Lemma 2.3 we conclude $f = g/\|g\|^2$, i.e., $\tilde{\alpha}(\xi) = C \cdot \sqrt{-2U(\xi)}$ and therefore $\alpha(t) = C \cdot \dot{x}(t)$ for some constant $C$ as $\alpha \cdot \dot{x} \geq 0$.

## A.2. Details on Two-Phase Trajectories

Let us discuss the different cases as introduced in Section 2 in further detail. We first ignore the constraint $t_* \leq t_{\max}$. The two phases are formed as follows:

1. Phase 1 consists of the first $k-1$ strokes. An excessive velocity $\dot{x}$ when reaching the left wall is eliminated by the inelastic bump. Hence, the minimal loss is obtained by choosing $C_{1,k}$ just as small such that we reach $x_{\min}$ at zero velocity while maintaining $k-1$ strokes in phase 1.

2. In phase 2 we look for the smallest $C_{2,k}$ such that we reach $x_*$ at velocity $v_*$ from state $(x_{\min}, 0)$ in a single stroke. That is, the trajectory in phase 2 is actually independent of $k$ (still assuming $t_* \leq t_{\max}$).

Let us denote by $v_{\text{wall}}$ the velocity at which we hit the left wall at $x_{\text{min}}$. So far we discussed $v_{\text{wall}} = 0$. The larger $k$ becomes the larger becomes $t_*$. At some $k$ we would violate $t_* \leq t_{\text{max}}$. Roughly speaking, if we increase $C_{1,k}$ and $C_{2,k}$ then we might reach $x_*$ at $t_* \leq t_{\text{max}}$ again, but $v_{\text{wall}} \neq 0$ when we increase $C_{1,k}$. However, note that a given $v_{\text{wall}}$ determines the $\ell$-optimal $C_{1,k}, C_{2,k}$ in order to reach $t_* = t_{\text{max}}$ in this case: $C_{1,k}$ is chosen small enough to reach the left wall with velocity $v_{\text{wall}}$. This determines the time $\tau$ at which the wall is hit. Then we choose $C_{2,k}$ small enough to reach $x_*$ in time $t_{\text{max}} - \tau$ for phase 2. Hence, when $k$ is large enough such that reaching $x_*$ at velocity $v_*$ would violate $t_* \leq t_{\text{max}}$, we have to search for the $\ell$-optimal $v_{\text{wall}}$.

To sum up, for each $k$ we can determine the $\ell$-optimal $C_{1,k}$ and $C_{2,k}$, yielding the optimal $\ell_k$. Then we exhaustively test all $k$ to find the optimum.

### A.3. Optimal Policy for A-Priori Known Starting Location

When the start position $x_0$ is given a priori, this knowledge can be exploited to improve the optimal control strategy $\pi_{\text{ana}}$ further; we denote the resulting strategy by $\pi_{\text{opt},x_0}$. Figure 6 shows $\pi_{\text{opt},x_0}$ with $x_0 = -0.55$. We can see that, contrary to $\pi_{\text{ana}}$ depicted in Figure 5, its trajectory reaches the left wall with a velocity close to zero, preventing loss of excessive velocity due to the inelastic bump. We see that the bootstrapping action $\alpha_{\text{boot}}(x, v) = 0.1$ for $|x - \hat{x}| \leq 0.01$ extends to both sides of the plane from the center, showing small impact given that only a little fraction of the state space is affected.

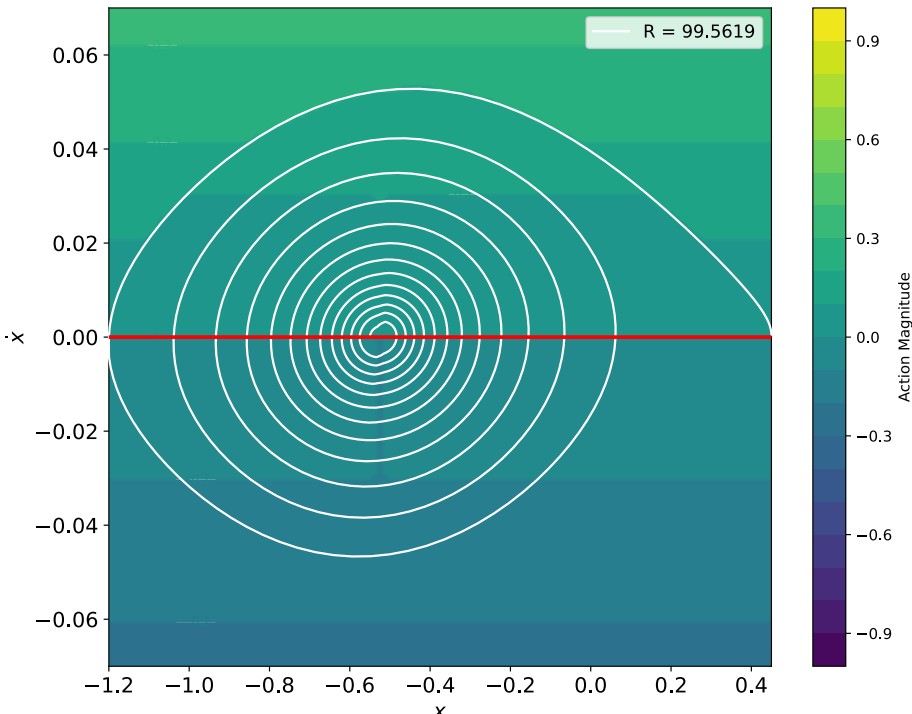

*Figure 6.* Optimal policy $\pi_{\text{opt},x_0}$ for $x_0 = -0.55$ with $C_{1,k} = 3.2986$ and $C_{2,k} = 4.8358$, with $k = 25$.

### A.4. Unconstrained Experiments

Recall our discussion after Theorem 2.4: When we have a small $C$ then little action is applied, so for each stroke the potential $U$ is slowly lowered and the number $k$ of strokes will be high until the goal is reached. When $C$ is increased the number $k$ is reduced in discrete steps and so is $\xi_* - \xi_0$.

While $k$ stays constant, however, increasing $C$ slightly increases $\xi_* - \xi$ as each stroke becomes slightly longer. See more details in Section A.4.

We move the left wall to $x_{\text{min}} = -\frac{2\pi}{6} - 0.45$ to verify single-phase trajectories for rising values of the parameter $C$. Figure 7 shows the instance $x_0 = -0.55$. As we increase $C$, $x_*$ is reached at increased speed $\dot{x}(t_*)$, increasing $\ell$. Jumps indicate a change in the stroke count $k$. If $\min_t x(t) = x_{k-1}$ becomes less than $x_{\text{min}}$ after a sufficiently large $k$ then these single-phase

solutions are not feasible.

Conversely, the position $x_{\min}$ of the left wall is feasible if and only if the car can reach the goal in one stroke at $\alpha = 1$. If the left wall is approximately in the interval $[-0.651, 0.310]$ then this problem is infeasible, and otherwise the Mountain Car can indeed reach the goal (from any start position).

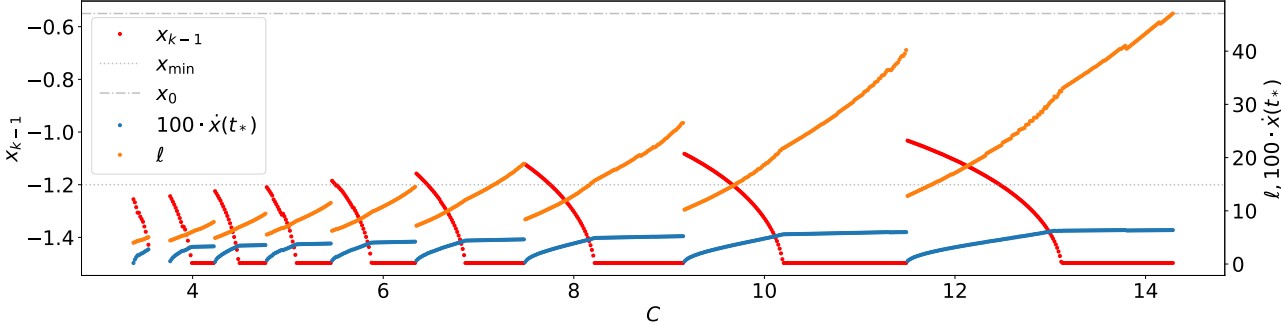

*Figure 7.* Loss $\ell$ over $C$ for $x_0 = -0.55$, together with $x_{k-1}$ and $\dot{x}(t_*)$ in the unconstrained setting.

### A.5. Optimality of the Discrete Control Problem

To confirm that the continuous-time analytical policy is also the optimal solution of the discrete-time case, the optimization problem

$$
\begin{aligned}
\min_{\alpha_i, \, i^*} \quad & \sum_{i=0}^{i^*-1} \alpha_i^2 \\
\text{s.t.} \quad & x_{i^*} \geq 0.45, \\
& |v_i| \leq 0.07, \\
& x_i \geq -1.2, \\
& |\alpha_i| \leq 1, \\
& i^* \leq 999
\end{aligned}
\tag{14}
$$

was solved for the three starting points $x_0 = -0.6$, $x_0 = -\pi/6$ and $x_0 = -0.4$. The obtained return values only marginally differ in the third decimal compared to the two-phase analytic solution. Since the numerically obtained action sequence could be related to a local minimum of the objective – particularly due to the large number of optimization variables, the guarantee that a global minimum was found is not given – , another test was executed. The optimization problem was simplified by only considering a single stroke, which reduces the number of optimization variables significantly. Starting from $x_0 = -0.6$ and $x_0 = -\pi/6 - 0.001$, the policy from Theorem 2.4 was applied to the discrete-time simulation model with an arbitrarily chosen $C = 20$. Using this policy, the car will execute one stroke until it reaches zero velocity at position $x_e$. This position is then used as target position $x_i^*$ in the optimization problem (14). This time, the initial guess of the action sequence is zero, i.e., the solver is not provided with any solution and now it should still find the same action sequence as given by Theorem 2.4.

Indeed, the optimizer could slightly improve the loss compared to the given policy ($x_0 = -0.6$: $0.5835$ analytic vs. $0.57246$ optimized; $x_0 = -\pi/6 - 0.001$: $9.983 \cdot 10^{-5}$ analytic vs. $9.772 \cdot 10^{-5}$ optimized). However, the shape of the action sequence is very similar to the analytically derived policy. To get an idea of the effect of discretization, in addition to the numerical approximation of the integral of the squared action over time, equation (6) was numerically evaluated, i.e., the loss was computed by integration along the position. The observed discrepancy between the two approximations of the integral was larger than the discrepancies observed between analytical solution of the continuous time problem compared to the numerical solution of the discrete-time problem. Therefore, we conclude that the analytically derived policy for the continuous time case also holds for the discrete-time case, since the observed differences are smaller than effects originating from the time discretization.

# B. Details on Chebyshev Policies

## B.1. Multi-Variate Horner Scheme

Let $p(x) = a_0 + a_1 x + \cdots + a_d x^d$ denote a polynomial of degree $d$ then the classic Horner scheme factors the $x$ as follows:

$$p(x) = \underbrace{\left( \cdots \left( (a_d \cdot x + a_{d-1}) \cdot x + a_{d-2} \right) \cdots + a_1 \right) \cdot x}_{d \text{ parentheses}} + a_0 \tag{15}$$

The number of multiplications and additions used is $d$, which is the smallest possible. There is varying literature on multi-variate generalizations. Since we use the notion of max-degree in this paper, we briefly provide an analysis for the number of multiplications and additions for $n$-variate max-degree $d$ polynomials here.

We see an $n$-variate polynomial as a polynomial of polynomials of polynomials and so on, each nesting level corresponding to one variable. Hence, we apply the Horner scheme per variable, which is per nesting level. That is, in Equation (15), when $p$ is an $n$-variate polynomial of max-degree $d$ then $a_i$ are $(n-1)$-variate polynomials, also of max-degree $d$.

**Lemma B.1.** *Evaluating an $n$-variate max-degree $d$ polynomial with the multi-variate Horner scheme leads to $(d+1)^n - 1$ multiplications and additions.*

*Proof by induction.* The case $n = 1$ is given in Equation (15). For the induction step $n \to n + 1$ observe that the $a_0, \ldots, a_d$ in Equation (15) each take by assumption $(d+1)^n - 1$ multiplications and additions, so we have in total $((d+1)^n - 1) \cdot (d+1) + d = (d+1)^{n+1} - 1$ multiplications and additions. $\square$

Note that such a polynomial has $(d+1)^n$ monomials.

# C. Details on Mountain Car Experiments

## C.1. Training and Evaluation Protocol

REINFORCE training was conducted with AdamW optimizer and a discount factor of $0.9$, using the following reproducible protocol: We trained 20 policies for 100 episodes. In each episode the environment picks $x_0 \sim \mathcal{U}([-0.6, -0.4])$. Then $\sigma(s)$ in the stochastic policy was set zero (for a deterministic evaluation) and 50 evaluation episodes were conducted. The one policy with the highest average return is picked for evaluation.

We did this for Chebyshev policies of degree 3. (Degree 3 performs slightly better than higher degrees, e.g., degree 5.) We call the resulting policy CH-3-REI. In Section C.2 we give further details using other optimizers than AdamW. With CH-3-ARS and CH-3-PPO we also trained 20 policies each, with 80 k steps for the former and 70 k steps for the latter. We then took pretrained RL Baselines3 Zoo agents on the MountainCarContinuous-v0 problem, see (Raffin et al., 2024), namely for ARS, SAC, PPO, Deep Deterministic Policy Gradient (DDPG), Twin Delayed Deep Deterministic Policy Gradient (td3), Advantage Actor-Critic (A2C), and TRPO.

Each of the agents together with $\pi_{\text{ana}}$ were evaluated as follows: We ran them with $x_0$ chosen at 100 evenly spaced points in $[-0.6, -0.4]$ and recorded the achieved return $R$. The mean return $\overline{R}$ is a faithful estimator for the expected return $\mathbb{E}_{x_0 \sim \mathcal{U}([-0.6, -0.4])}(R)$. The three best performing SOTA agents turned out to be ARS, SAC and PPO, as summarized in Table 1. A full comparison is given in Section C.5. The regret $r$ is the difference of $\overline{R}$ of the evaluated policy compared to the one of $\pi_{\text{ana}}$. In particular, CH-3-REI trained by classical REINFORCE improves the regret $(0.77)$ compared to ARS $(2.72)$, SAC $(4.78)$ and PPO $(5.48)$ by a factor of 3.5, 6.2 and 7.1, respectively, which is surprising given that REINFORCE as the first policy-gradient algorithm is considered clearly inferior to modern algorithms like PPO or SAC.

All experiments were conducted on an Intel Core i7-7800X CPU with 32 GB of RAM. The most time consuming task was the training protocol for CH-3-REI which took $90 \, \text{min}$ with the evaluation taking another $30 \, \text{min}$. Less time is required for training and evaluation of CH-ARS and CH-PPO as well as evaluating the RL Baselines3 Zoo agents.

## C.2. Training Details for REINFORCE with Different Optimizers

Given the simple nature of REINFORCE as the first policy-gradient RL algorithm, we took a deeper look on the effect of different optimizers on the training of Chebyshev policies. (This is partly motivated by investigations of (Waclawek &

Huber, 2024) on uni-variate piecewise Chebyshev polynomials in supervised learning tasks, where the effect of different optimizers was pronounced.)

We reimplemented the classical Monte Carlo policy gradient REINFORCE algorithm according to (Sutton & Barto, 2018), as it is not part of PyTorch. We trained Chebyshev policies of max-degree 3 using the classical REINFORCE algorithm according to (Sutton & Barto, 2018), utilizing gradient-based PyTorch optimizers. As in Section C.1 20 policies were trained for 100 episodes, followed by 50 evaluation episodes. The results of the evaluation experiment for CH-3-REI are depicted in Figure 8. As it is typical for REINFORCE, we have a large spread in performance of the trained policies.

The one policy with the highest average return was selected and is denoted by CH-3-REI, which stems from the AdamW optimizer. The step size for training was set to $0.0003$ and $\theta_{i_1,\dots,i_n}^{(\sigma)}$ where initialized s.t. $\sigma(s)$ is a value close to 1, enabling initial exploration. $\sigma(s) = 0$ during evaluation. Training of policies using SGD or L-BFGS optimizers diverged.

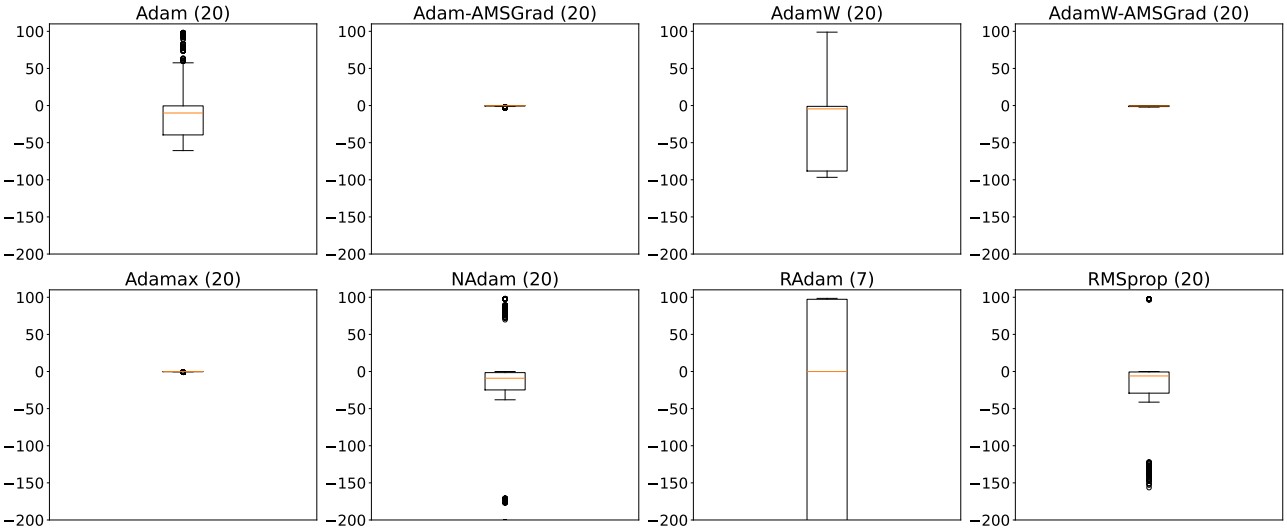

*Figure 8.* Mountain Car: Evaluation of CH-3-REI results trained with different optimizers, number of non-diverging policies utilized is shown in brackets (20 before training). 50 episodes per policy, each datapoint is the return of one episode.

### C.3. PPO with Different MLP Architectures

A pressing research question from our analysis in Section 3 concerns the surprisingly high regret of MLP policies with SOTA RL algorithms, like SAC, PPO or ARS alike.

Let us focus on PPO, which is known for its wide applicability, and ask whether the default network size $[64, 64]$ of the MLP policy would explain the impaired performance. We can exclude underfitting from the simplicity of the optimal policy $\pi_{\text{ana}}$. Hence, we can focus on a possible overfitting, although $[64, 64]$ is not quite large.

In Table 3 we reported on the achieved results for reduced number of layers and reduced sizes of layers. To sum up, when the network size is reduced, the agent's performance becomes even more degraded. The results of the default setting match the one reported on RL Baselines3 Zoo.

### C.4. REINFORCE Does Not Succeed with MLP Policies

A surprising result of Section 5 was that CH-3-REI based on REINFORCE performs significantly better than all MLP policies trained by different RL algorithms, while REINFORCE would not be able to actually train a successful MLP policy. Here we report on the details on the latter claim.

We again repeat the same training protocol as in Section C.1: Recall that a stochastic policy $\pi$ maps a state $s$ to a pair $(\mu(s), \sigma(s))$, which are here modeled by a MLP, respectively. The $\mu$-net is initialized by default initialization of PyTorch. The $\sigma$-net is initialized with small random numbers, only the last layer receives a bias to $0.25$. That is, $\sigma$ as a map initially gives a constant close to 1 over the state space with small random fluctuations. (This identical to what we did for Chebyshev policies.)

*Table 3.* Performance on Mountain Car. Mean, min and max return for PPO agents using different MLP architectures. Evaluated on 20 evenly spaced starting positions $x_0 \in [-0.6, -0.4]$.

| Layer Dimensions | $\overline{R}$ | $R_{\min}$ | $R_{\max}$ |
|---|---|---|---|
| $[64, 64]$ (original) | 94.22 | 91.16 | 95.15 |
| $[32, 32]$ | 92.60 | 89.02 | 95.82 |
| $[64]$ | 90.97 | 90.03 | 95.13 |
| $[16, 16]$ | $-0.22$ | $-0.32$ | $-0.16$ |
| $[16]$ | $-0.55$ | $-0.55$ | $-0.53$ |
| $[32]$ | $-5.34$ | $-7.80$ | $-4.02$ |

We tested various MLP architectures, starting with the default 2-layer architecture $[64, 64]$, but also smaller single- and two-layer variants. (Recall that the optimal policy $\pi_{\text{ana}}$ is quite simple, so $[64, 64]$ should by far suffice in terms of model capacity.)

In Figure 9, we report on the results for the six MLP architectures and eight different optimizers, and for each combination we trained 20 agents, resulting in a total of 960 training attempts. None of these solved the Mountain Car task, only one agent (RAdam, network size $[8]$) at times reaches the goal with a mean return of about 60.

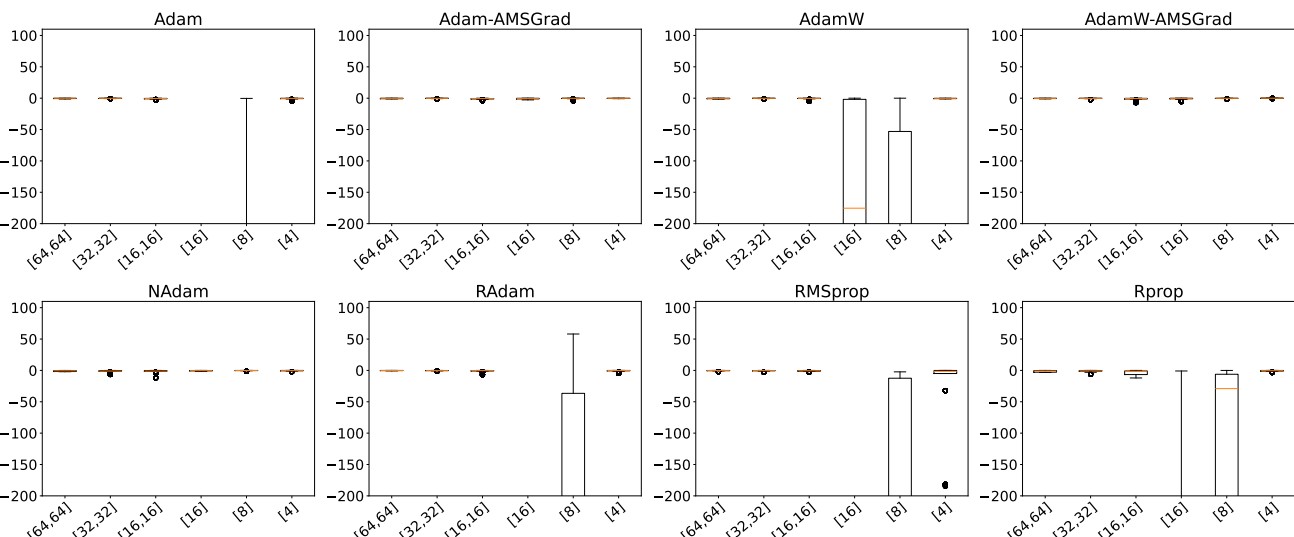

*Figure 9.* Mountain Car: Evaluation of MLP REINFORCE results trained with different optimizers. 50 episodes per policy, each datapoint is the return of one episode.

## C.5. Full Comparison of Chebyshev Policies Against All RL Baseline3 Zoo Agents

In the following take a deeper look on the results briefly summarized in Table 1 and present a full version of this table in Table 4. We compare all pretrained RL Baselines3 Zoo agents, which are in decreasing order of performance: ARS, SAC, PPO, DDPG, td3, TRPO, A2C and Truncated Quantile Critics (TQC). In Table 4 we leave out the TQC agent as its performance lies behind substantially ($\overline{R} = 61.14$ for $x_0 \in [-0.6, -0.4]$). In the following discussion a couple of aspects of the presented results.

First of all, we see that the mean target velocity $\overline{v_*}$ of the optimal policy is essentially zero, independent of $x_0$. This is what we expect from optimality, otherwise we would suffer from excess kinetic energy at the goal. Furthermore, $\pi_{\text{opt},x_0}$ essentially exploits the maximum possible time $t_*$ to reach the goal, with the remainder to 999 being explained by the restriction that the number of strokes is simply an integral number. In contrast to $\pi_{\text{opt},x_0}$, the analytical optimal policy $\pi_{\text{ana}}$ does not have the advantage of known $x_0$ a priori and hence has lower $\overline{t_*}$.

As already discussed in Table 1, all Chebyshev policies outperform all the MLP policies. Moreover, when we look at the standard deviation, we observe that the Chebyshev policies have lower standard deviation than MLP policies, except the

*Table 4.* Performance on Mountain Car. Mean, standard deviation, min, and max return $R$, and the same for velocity $v_*$ at target and episode length $t_*$ for different trained agents over 100 evenly spaced $x_0 \in [-0.6, -0.4]$.

| Policy | $\overline{R}$ | std($R$) | $R_{\min}$ | $R_{\max}$ | $\overline{v_*}$ | std($v_*$) | $v_{*,\min}$ | $v_{*,\max}$ | $\overline{t_*}$ | std($t_*$) | $t_{*,\min}$ | $t_{*,\max}$ | $\|\cdot\|_2$ |
|---|---|---|---|---|---|---|---|---|---|---|---|---|---|
| $\pi_{\mathrm{opt},x_0}$ | 99.59 | 0.0463 | 99.47 | 99.68 | $4.7 \cdot 10^{-4}$ | 0.0000 | $4.7 \cdot 10^{-4}$ | $4.7 \cdot 10^{-4}$ | 955.14 | 30.33 | 849 | 999 | |
| $\pi_{\mathrm{ana}}$ | 99.39 | 0.0768 | 99.15 | 99.52 | $4.7 \cdot 10^{-4}$ | 0.0000 | $4.7 \cdot 10^{-4}$ | $4.7 \cdot 10^{-4}$ | 769.25 | 90.8347 | 571 | 968 | |
| CH-3-ARS | 98.74 | 0.0987 | 98.95 | 99.11 | $1.8 \cdot 10^{-2}$ | 0.0032 | $7.2 \cdot 10^{-3}$ | $2.0 \cdot 10^{-2}$ | 471.65 | 125.6362 | 325 | 985 | 0.1523 |
| CH-3-REI | 98.62 | 0.1661 | 98.31 | 98.89 | $2.4 \cdot 10^{-2}$ | 0.0069 | $3.0 \cdot 10^{-3}$ | $2.8 \cdot 10^{-2}$ | 396.46 | 109.5182 | 246 | 838 | 0.0680 |
| CH-3-PPO | 98.10 | 0.2217 | 97.61 | 98.42 | $2.3 \cdot 10^{-2}$ | 0.0053 | $4.8 \cdot 10^{-3}$ | $2.6 \cdot 10^{-2}$ | 469.97 | 128.3894 | 334 | 985 | 0.0865 |
| ARS | 96.67 | 0.8562 | 92.51 | 97.42 | $4.2 \cdot 10^{-2}$ | 0.0130 | $1.1 \cdot 10^{-2}$ | $5.3 \cdot 10^{-2}$ | 239.28 | 84.1711 | 152 | 610 | 0.2105 |
| SAC | 94.61 | 1.2345 | 89.70 | 95.77 | $3.8 \cdot 10^{-2}$ | 0.0150 | $1.2 \cdot 10^{-2}$ | $6.1 \cdot 10^{-2}$ | 105.77 | 33.1034 | 76 | 179 | 0.3173 |
| PPO | 93.91 | 1.1693 | 90.86 | 95.23 | $3.4 \cdot 10^{-2}$ | 0.0066 | $5.7 \cdot 10^{-3}$ | $3.7 \cdot 10^{-2}$ | 298.01 | 93.6045 | 202 | 858 | 0.2730 |
| DDPG | 93.51 | 0.0486 | 93.43 | 93.56 | $3.9 \cdot 10^{-2}$ | 0.0048 | $2.9 \cdot 10^{-2}$ | $4.6 \cdot 10^{-2}$ | 66.37 | 0.4828 | 66 | 67 | 0.4267 |
| TD3 | 93.48 | 0.0772 | 93.36 | 93.62 | $3.4 \cdot 10^{-2}$ | 0.0022 | $2.9 \cdot 10^{-2}$ | $3.8 \cdot 10^{-2}$ | 65.95 | 0.7794 | 65 | 67 | 0.4260 |
| TRPO | 92.49 | 0.3872 | 90.16 | 92.78 | $5.2 \cdot 10^{-2}$ | 0.0078 | $3.7 \cdot 10^{-2}$ | $6.3 \cdot 10^{-2}$ | 86.84 | 9.5433 | 81 | 157 | 0.4008 |
| A2C | 91.16 | 0.2555 | 90.46 | 91.60 | $4.8 \cdot 10^{-2}$ | 0.0085 | $3.2 \cdot 10^{-2}$ | $6.0 \cdot 10^{-2}$ | 90.44 | 2.6583 | 86 | 98 | 0.4194 |

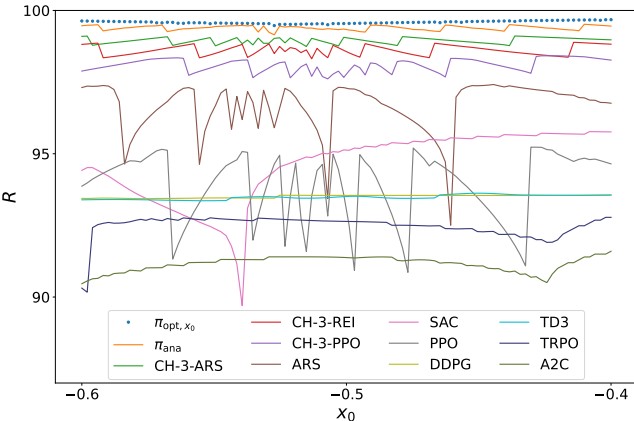

*Figure 10.* The return $R$ over start positions $x_0$ for all policies from Table 4.

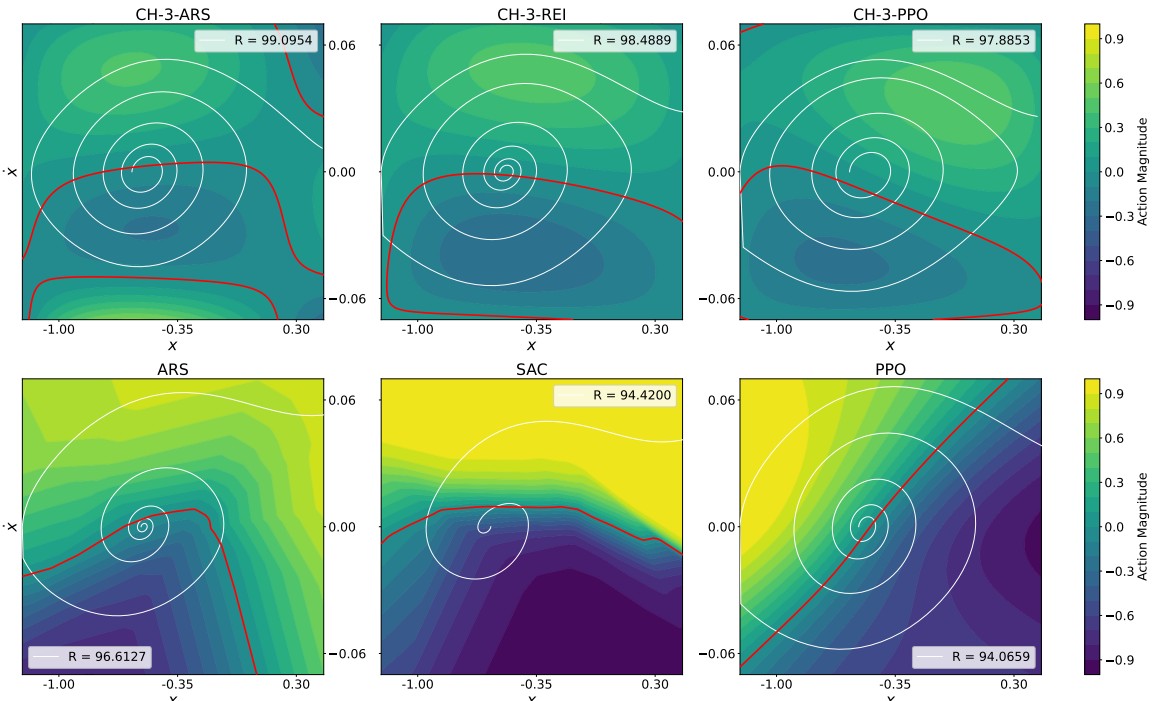

*Figure 11.* The figures plot the actions of Chebyshev agents and the three top-performing neural agents over the state space, the zero-actions in red and in white a trajectory from $x_0 = -0.55$.

low-performing MLP policies DDPG or worse.

Taking a closer look at the minimum return $R_{\min}$, we see that the top MLP policy ARS delivers a very inconsistent performance, at times falling below low-performing MLP policies, like DDPG or td3. We highlight this behavior in Figure 10, where we plot the return $R$ over all start positions $x_0$.

When we look at the mean time $\overline{t_*}$ when a policy reaches the goal, we consistently see that the Chebyshev policies exploit $t_* \leq 999$ more effectively an come closer to $\pi_{\mathrm{ana}}$. What we also observe is that the low-performing policies have a low standard deviation $\mathrm{std}(t_*)$, especially for DDPG to A2C. This indicates that they learn a policy leading to trajectories of equal length, quite independent of the start position $x_0$, which is consistent with low values of $\overline{t_*}$ in total, i.e., they reach the goal without exploiting multiple forth-and-back oscillations. This observation is confirmed by the plotted trajectory for SAC in Figure 11.

Furthermore, when we compare a policy $\pi$ to $\pi_{\mathrm{ana}}$ in terms of the $L_2$-norm, i.e., by looking at $\|\pi - \pi_{\mathrm{ana}}\|_2$ over the state space domain $[-1.2, 0.45] \times [-0.07, 0.07]$, then again the Chebyshev policies approximate $\pi_{\mathrm{ana}}$ much closer than the MLP policies. Taking a closer look at the CH-3-ARS in Figure 11, we see that the difference to $\pi_{\mathrm{ana}}$ mainly occurs in areas with no or less relevance for the occurring trajectories.

### C.6. In-Depth Comparison of Policies

In Figure 11 we extend our discussion from Figure 5 by extending the policy plots to all Chebyshev policies and the three top performing MLP policies. The observations we mentioned in Section 5 generalize to the larger set of policies in Figure 11 as follows.

All MLP policies generate larger actions than Chebyshev policies. Consequently, the number of strokes for MLP is smaller, so is $t_*$, which lowers the return. Only PPO displays a larger number of strokes, but only because it counteracts the dynamics of Mountain Car, i.e., it outputs actions in opposite direction of the velocity of the car.

This counter action of the dynamics is also present for ARS, though at regions of the state space with less impact on the trajectories. This also illustrates limitations of the quality criterion $\|\pi - \pi_{\mathrm{ana}}\|_2$ when the deviation is mainly attributed to irrelevant regions of the state space.

### C.7. Curse of Dimensionality

For a state space of dimension $n$ and max-degree $d$, we have $(d+1)^n$ Chebyshev polynomials. To investigate the impact of $d$ on the return, computational performance, and numerical stability, we repeat training of Chebyshev policies using PPO on Mountain Car with increasing max-degree from 1 up to 50. For each degree, again, 20 agents were trained independently and evaluated by averaging the return over 100 deterministic starting states sampled from the initialization range. The results are shown in Table 5.

Overall, return improves rapidly once the policy class becomes sufficiently expressive. Moreover, the return remains stable over a wide range of higher degrees, still beating all neural SOTA agents even with $d = 20$. Beginning with degrees around 30, however, several agents do not succeed in reaching the goal flag any longer and standard deviation of overall return increases.

However, these results highly depend on the choice of RL algorithm. Classical REINFORCE would already fail at max-degree 10. Also, performance (i.e., simulation steps per second) drops, though not quadratic (as the number of polynomials). However, we have not optimized for computational performance yet, e.g., employ dynamic programming strategies on the recursive computation scheme of multi-variate Chebyshev polynomials.

### C.8. Evaluation with Additional Stable Baselines3 RL Algorithms

In the following, we extend our experimental evaluation to different RL algorithms available in Stable Baselines3. However, in order to avoid laborous native implementation of Chebyshev policies for different RL algorithms, we followed the following approach: We use a single neuron without bias and linear activation and pass it as network architecture to the Stable Baselines3 API, to effectively form the linear combination of the Chebyshev polynomials as in Equation (10).

We again run experiments on Mountain Car. To verify consistency, we also re-ran PPO and ARS experiments with this setup, and added SAC as additional algorithm. In this way, results now cover an on-policy, off-policy and random search

*Table 5.* CH-3-PPO performance on Mountain Car for different polynomial degrees. Training was conducted with 20 agents for each degree, showing mean return $\overline{R}$ and standard deviation across all agents as well as the best performing agent per degree. Evaluation conducted over 100 evenly spaced $x_0 \in [-0.6, -0.4]$.

| Degree | $\overline{R}$ all | $\text{std}(R)$ all | $\overline{R}$ best | $\text{std}(R)$ best | Steps/s |
|---|---|---|---|---|---|
| 1 | -0.0975 | 0.0842 | -0.0043 | 0.0002 | 61.0 |
| 2 | -0.6159 | 0.1649 | -0.3784 | 0.2569 | 53.9 |
| 3 | 88.4433 | 30.117 | 98.9623 | 0.0769 | 50.4 |
| 4 | 97.9296 | 0.1271 | 98.1458 | 0.2278 | 46.9 |
| 5 | 97.78 | 0.1565 | 98.2048 | 0.1625 | 43.0 |
| 6 | 97.5848 | 0.1891 | 98.0292 | 0.1855 | 38.5 |
| 7 | 97.6631 | 0.2513 | 98.11 | 0.2196 | 36.9 |
| 8 | 97.5722 | 0.3585 | 98.1088 | 0.2061 | 34.5 |
| 9 | 97.623 | 0.244 | 97.9115 | 0.2099 | 32.0 |
| 10 | 97.3698 | 0.2612 | 97.7654 | 0.2519 | 29.2 |
| 11 | 97.4387 | 0.3251 | 97.838 | 0.2133 | 27.8 |
| 12 | 97.3455 | 0.2943 | 97.8038 | 0.2306 | 25.6 |
| 13 | 97.3748 | 0.2645 | 97.8012 | 0.1816 | 23.2 |
| 14 | 97.2876 | 0.2562 | 97.6679 | 0.1562 | 22.8 |
| 15 | 97.1798 | 0.3041 | 97.6372 | 0.3292 | 21.9 |
| 16 | 97.0765 | 0.5233 | 97.7306 | 0.1682 | 19.4 |
| 17 | 97.0275 | 0.3151 | 97.5145 | 0.1303 | 18.6 |
| 18 | 96.952 | 0.2965 | 97.4589 | 0.1814 | 18.0 |
| 19 | 96.6435 | 0.4085 | 97.3213 | 0.185 | 16.1 |
| 20 | 96.6991 | 0.7013 | 97.3904 | 0.1901 | 15.4 |
| 30 | 77.2577 | 38.7163 | 95.7942 | 0.331 | 9.8 |
| 40 | 80.4135 | 19.5194 | 94.6989 | 0.4354 | 6.0 |
| 50 | 80.7739 | 21.937 | 94.9583 | 0.5967 | 4.0 |

*Table 6.* Performance on Mountain Car with Chebyshev representation layer with single-node MLP architecture. Training 20 agents with distinct seeds and picking the result with the highest mean return $\overline{R}$. Showing mean return $\overline{R}$ (regret $r$), standard deviation as well as min and max return.

| Pol. $\pi$ | $\overline{R} \uparrow (r \downarrow)$ | $\text{std}(R)$ | $\min R - \max R$ |
|---|---|---|---|
| $\pi_{\text{ana}}$ | 99.39 | 0.0768 | 99.15 – 99.52 |
| **CH-3-ARS** | 98.72 (0.67) | 0.13 | 98.45–98.93 |
| **CH-4-PPO** | 98.47 (0.92) | 0.16 | 98.17–98.74 |
| **CH-3-SAC** | 98.19 (1.20) | 0.22 | 97.64–98.52 |

algorithm. Results (20 independently trained agents, evaluated over 100 uniform initial states) are shown in Table 6. We see that both ARS and PPO results align with what we have seen with native implementations (see Table 4). CH-SAC follows this line of results, still outperforming all MLP SOTA agents (including SAC with MLP).

# D. Experimental Results on Additional Environments

### D.1. Pendulum

In this section we extend our discussion from Section 5 concerning the performance of Chebyshev policies against pretrained RL Baselines3 Zoo agents on the Gymnasium environment *Pendulum-v1*. For the sake of comparison to other environments, we again trained Chebyshev policies with ARS, PPO and REINFORCE. Preliminary experiments revealed that ARS performs best with max-degree 6 Chebyshev policies, PPO performs best with max-degree 5 and REINFORCE performs best with max-degree 3, and they are accordingly named CH-6-ARS, CH-5-PPO, and so on. In the following we exclude CH-3-REI as it lags behind significantly with $\overline{R} = -466.29$.

We summarize our experimental results in Table 7. As we discussed in Section 5, Chebyshev approximators improve the mean return of MLP policies for ARS and PPO. We also list the standard deviation $\text{std}(R)$ in Table 7 and see that the attained returns are quite spread, which is expected given that a random initialization of the pendulum in upright position enables a possible return of zero, while the opposite initial position lowers the return substantially. (The optimal control for this environment is unknown, and hence we can only speculate on the regret of the current SOTA.)

The high standard deviation motivated us to take a closer look at the density function of the return distribution, see Figure 12. Here we see that indeed the Chebyshev policies move density mass from lower returns to higher returns and, moreover, the performance of the top-performer SAC and CH6-ARS are very close, what we also see from Table 7. Interestingly, for MLP policies, ARS actually performed worst.

### D.2. Real-World Quanser Aero 2 System

**Mechatronic Background** The Quanser Aero 2 is a motion control testbed for multiple-input-multiple-output control theory with the goal to position the beam by actuating two fans, see Figure 13. It can be put into a 1-DOF or 2-DOF configuration. In either case it displays pronounced non-linear system behavior by design for educational reasons.

*Table 7.* Performance on pendulum environment. Evaluation over $50 \times 50$ evenly spaced initial angles from $[-\pi, \pi]$ and angular velocities from $[-1, 1]$, showing mean, standard deviation, min and max returns.

| Policy | $\overline{R}$ | $\text{std}(R)$ | $\min R$ | $\max R$ |
|---|---|---|---|---|
| SAC | -147.24 | 83.39 | -377.46 | -1.08 |
| TQC | -147.83 | 84.22 | -380.56 | -1.01 |
| **CH-6-ARS** | -150.80 | 88.12 | -392.02 | -0.21 |
| TD3 | -155.00 | 92.71 | -377.14 | -0.51 |
| DDPG | -156.64 | 106.50 | -1491.25 | -1.26 |
| **CH-5-PPO** | -162.75 | 98.62 | -486.60 | -1.97 |
| A2C | -167.90 | 102.09 | -528.22 | -0.08 |
| **PPO** | -176.16 | 107.17 | -516.15 | -3.83 |
| TRPO | -180.65 | 141.49 | -1491.89 | -0.25 |
| **ARS** | -218.32 | 159.30 | -784.63 | -0.07 |

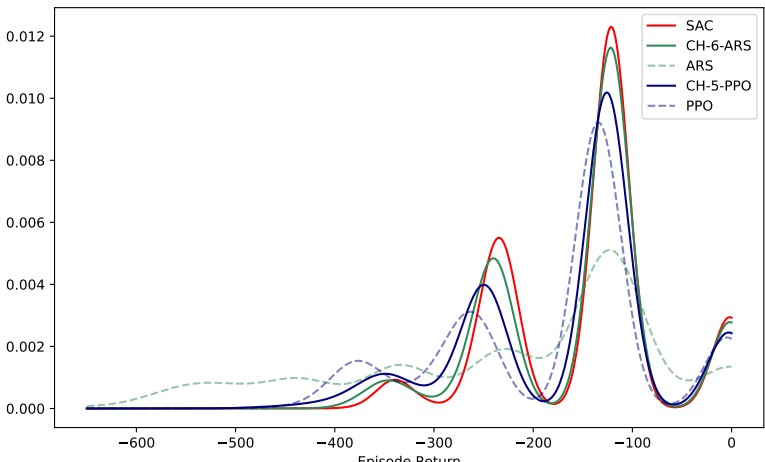

*Figure 12.* Pendulum: Density function of the return distribution for Chebyshev policies and their MLP counterparts along with the top-performing policy trained by SAC.

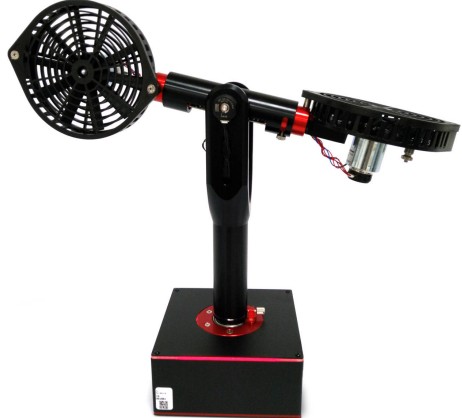

*Figure 13.* The Quanser Aero 2 testbed in 2-DOF configuration. We turn the fans in the same direction and lock the main arm in order to obtain the 1-DOF configuration of the Aero 2.

We follow the configuration in (Schäfer et al., 2024a), which is the 1-DOF configuration. In this work, a control task is set up where the beam shall follow a target pitch signal. The RL agent observes the actual pitch angle, actual angular velocity and a target pitch angle, and outputs actions in form of a continuous voltage for the fans. The state at time $t$ is formalized as a vector $(\theta_t, \theta^*, \theta_t - \theta_{t-1})$, where $\theta_t$ is the actual pitch angle, $\theta_t^*$ is the target pitch angle, and $\theta_t - \theta_{t-1}$ captures the actual pitch velocity, at time $t$, respectively. The action is a scalar output voltage. The objective is to minimize the absolute tracking error over time, formalized by the return $R = \sum_t r_t$, which is the cumulation of the reward $r_t = -|\theta_t - \theta_t^*|$.

**Environment Implementation Details**  For the RL training, we again leverage the Gymnasium environment framework and use Stable Baselines3 implementations of PPO and ARS. To this end, we needed to implemented a custom Gymnasium environment that integrates the Aero 2 system. This implementation uses a high-fidelity simulation model of the Aero 2 system dynamics as introduced and verified by (Schäfer et al., 2024b). The code for this environment is published at (Huber et al., 2026).

**Training and Evaluation**  Since the control task is to follow a target pitch signal, for our training, we generate a target pitch signal that (i) is continuous with limited dynamics such that an agent can potentially follow given the dynamic ability of the Aero 2 and (ii) also contains constant sections in order to judge the steady state error, which is an interesting quality criterion in control theory. (Note that return maximization implicitly leads to stead-state error minimization.)

To this end, we generate for each episode a target pitch signals that consist of a random, continuous sequence of sinusoidal

and constant sections. Its total length is 2000 steps, which equals to $200\,\mathrm{s}$ by a sample time of $100\,\mathrm{ms}$. The sections are of uniform random length between $2.25\,\mathrm{s}$ to $15\,\mathrm{s}$ and each section ends at a new intermediate uniform random target pitch in the interval $-40°$ to $40°$, i.e., $-0.698\,\mathrm{rad}$ to $0.698\,\mathrm{rad}$. In Figure 14 at the top subfigure we see such a random target pitch signal.

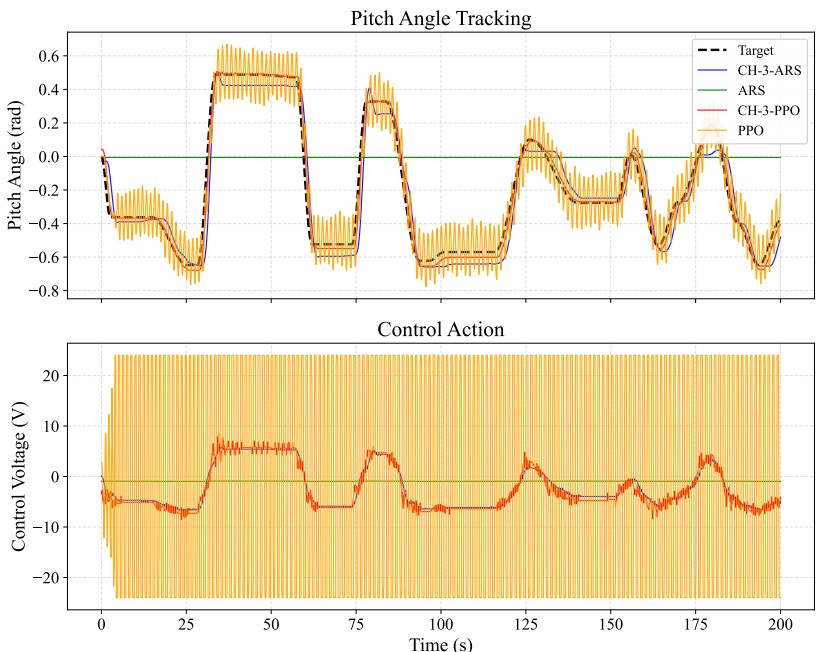

*Figure 14.* Details on one of ten evaluation trajectories performed on the real-world Quanser Aero 2 system.

We train MLP and Chebyshev policies with PPO and ARS as in the other experiments. For the MLP we again use the default 2-layer network architecture of size $[64, 64]$, as suggested in (Schäfer et al., 2024a). For the Chebyshev policies we found that max-degree 3 polynomials worked best. PPO trained for $150\,000$ steps and ARS trained for $4\,000\,000$, for both MLP and Chebyshev policies. As in previous experiments, we train 12 agents for each combination and select the best policy for evaluation.

We evaluate the policies on a set of 10 random target pitch signals and collect the return, the mean power consumption and the mean action magnitude. The seed of the evaluation set is fixed and therefore reproducible. The power consumption is calculated by the product of motor coil current and voltage, i.e., it is measured on the real system and simulated in Gymnasium. In Table 8 we summarize the results when the evaluation is performed in the simulation environment.

We furthermore evaluated the policies trained in simulation on the real-world system, which provides us with insights on the sim-to-real transferability. The results are summarized in Table 9.

The sim-to-real transferability is paramount for real-world RL, because from-scratch training of RL agents on the real system is typically prohibited due to safety limitations, risk of damage and wear, and is bound to real-time speed and therefore virtually unfeasible. So even if we continue training on the real-system, we typically have to start with agents pretrained in simulation.

Let us first discuss the results from the simulation environment in Table 8. In all cases, Chebyshev policies again outperform their MLP counterparts.

In more detail, we recognize that ARS with an MLP policy essentially failed to control the beam in a way to follow the target pitch signal sufficiently, leading to a high average pitch deviation (i.e., large negative return), see also further details in Figure 14. (Note that Figure 14 actually displays the results on the real system, but the main messages are the same for the simulation environment.) In contrast, ARS did succeed for Chebyshev policies, although CH-3-ARS clearly displays a significant steady-state error in Figure 14.

PPO achieves a much better pitch deviation than ARS. However, the output voltage (and power consumption) is remarkably high and a closer look to Figure 14 reveals the reason for this: PPO displays pronounced oscillation around the target pitch

caused by the bang-bang strategy apparently learned by PPO. However, again with Chebyshev policies, this unfavorable behavior disappears and CH-3-PPO achieves the best return and also displays most favorable dynamic behavior from a control-theoretic point of view, which we see in power consumption and action magnitude.

Finally, let us discuss the performance of the policies when evaluated on the real system. Since ARS failed to control the beam in simulation, it is expected that it also does so on the real system and hence we receive essentially the same pitch deviation. For CH-3-ARS we see that the pitch deviation gets worse by $31\%$, so it moderately retained its performance. We see that the transfer from simulation to the real system caused PPO to increase its pitch deviation by a factor of $2.151$, while CH-3-PPO essentially could retain its performance and only increased the pitch deviation by $13.4\%$. As a result, on the real system CH-3-PPO now outperforms PPO by a factor of $3.3$ in terms of pitch deviation. On a side note, the bang-bang control of PPO actually caused significant heat on the motors of the Aero 2, and required us to insert cool-down phases in the evaluation procedure.

*Table 8.* Performance evaluation on the Quanser Aero 2 Gymnasium simulation environment over 10 evaluation episodes. For each episode we report on the average pitch deviation, average action magnitude and average power consumption and list it as $\mu \pm \sigma$, where $\mu$ is the mean and $\sigma$ is the standard deviation over the 10 episodes. The mean return $\overline{R}$ over all episodes is the negative average pitch deviation multiplied by the number of episode steps, which is 2000.

| Pol. $\pi$ | Pitch deviation (rad) $\downarrow$ | $\overline{R} \uparrow$ | Action Magnitude (V) $\downarrow$ | Power Consumption (W) $\downarrow$ |
|---|---|---|---|---|
| CH-3-ARS | $0.0626 \pm 0.0113$ | $-125.2$ | $4.0828 \pm 0.4391$ | $1.3331 \pm 0.2465$ |
| ARS | $0.3609 \pm 0.0615$ | $-721.8$ | $0.9305 \pm 0.0063$ | $0.0559 \pm 0.0008$ |
| CH-3-PPO | $0.0246 \pm 0.0038$ | $-49.2$ | $4.4338 \pm 0.5889$ | $1.6545 \pm 0.3647$ |
| PPO | $0.0423 \pm 0.0037$ | $-84.6$ | $16.3623 \pm 0.4699$ | $37.0509 \pm 1.9717$ |

*Table 9.* Performance evaluation of agents trained in simulation on the real Quanser Aero 2 system without additional training in analogy to Table 8. We also listed the quotient of the pitch deviation between the evaluation on the real and the simulated system.

| Pol. $\pi$ | Pitch deviation (rad) $\downarrow$ | Real/sim deviation $\downarrow$ | $\overline{R} \uparrow$ | Action Magnitude (V) $\downarrow$ | Power Consumption (W) $\downarrow$ |
|---|---|---|---|---|---|
| CH-3-ARS | $0.0821 \pm 0.0064$ | $1.312$ | $-164.2$ | $4.1984 \pm 0.3757$ | $0.3604 \pm 0.0765$ |
| ARS | $0.3592 \pm 0.0512$ | $0.995$ | $-718.4$ | $0.9275 \pm 0.0067$ | $0.0049 \pm 0.0006$ |
| CH-3-PPO | $0.0279 \pm 0.0054$ | $1.134$ | $-55.8$ | $4.6035 \pm 0.3789$ | $0.4966 \pm 0.0857$ |
| PPO | $0.0910 \pm 0.0064$ | $2.151$ | $-182.0$ | $21.1870 \pm 0.9451$ | $52.6000 \pm 2.2246$ |

