# OpenReview forum: "Chebyshev Policies and the Mountain Car Problem: Reinforcement Learning for Low-Dimensional Control Tasks"
_ICML.cc/2026/Conference — ICML 2026 spotlight_

### Official Review · Reviewer_548M · 2026-03-05

**Soundness:** 4
**Presentation:** 3
**Significance:** 3
**Originality:** 3
**Overall Recommendation:** 5
**Confidence:** 4

**Summary:**

The paper proposes a closed-form analytical solution to the mountain car problem, yielding an optimal control policy. It further proposes replacing neural networks with Chebyshev Polynomials as parameterized policies. It shows Chebyshev policies outperforming MLP-based RL algorithms in the mountain car, pendulum, and a helicopter control task.

**Compliance With Llm Reviewing Policy:**

Affirmed.

**Final Justification:**

Solid paper, combining nice theory with good results. Can form the foundation of considering closed-form solutions for control problems in RL and for using Chebyshev policies in subsequent works. Well-written presentation that does not require significant changes to be understandable by a range of readers. In my opinion, should definitely be accepted.

**Key Questions For Authors:**

1. Experiments: what caused the choice of PPO, ARS, and REINFORCE as baselines compared to other algorithms like A2C or SAC?
2. Could you derive a feasibility bound for the mountain car problem, i.e., a minimum distance between $x_{min}$ and $x_*$? That might be a nice side result for the theoretical section of the paper.
3. How would a combination between MLPs and Chebyshev layers look like, which you suggested in the conclusion?

**Limitations:**

Yes

**Strengths And Weaknesses:**

Strengths
1. Major strengths \
1.1 Proposing the use of Chebyshev policies in RL and showing their performance potential is a strong contribution. So far, most works in RL use neural networks, which may not be ideal for all kinds of tasks. Chebyshev policies can be a good alternative for some control environments. The paper is honest about restricting Chebyshev policies to control problems, which is a sensible restriction. \
1.2 Showing how Chebyshev policies can easily be included in policy gradient RL algorithms is very beneficial to facilitate their subsequent adoption
2. Minor strengths \
2.1 Deriving an upper bound for the performance on the mountain car problem is beneficial to RL researchers using it as a test environment \
2.2 The experimental results cover several control environments, including a simple real-world robot. The consistently good performance of Chebyshev policies highlights is future potential.

Weaknesses
1. Major weaknesses \
1.1 The contribution appears diluted: the paper proposes both a theoretical result and a loosely related new policy architecture. While both are beneficial, this dilutes the contribution from my point of view. Chebyshev policies do not necessarily require the closed-form solution of the mountain car problem, while the analytical solution is valid without Chebyshev policies. The result that the optimal policy to the mountain car problem is simple is not overly surprising given the simplified dynamics of the environment, it is somewhat detached from proposing Chebyshev policies for several control tasks afterwards. I would encourage the authors to clarify their contribution and storyline to show what their intended main contribution is. \
1.2 The range of possible applications is somewhat narrow: Chebyshev policies are only proposed for low-dimensionality control tasks with continuous action spaces and rather simple environment dynamics. While relevant, this is a restrictive subset of environments. Addressing these issues would be out of scope for an ICML paper, but I would encourage the authors to address this in a bit more detail than currently in the paper.
2. Minor weaknesses \
2.1 Please introduce abbreviations properly, e.g., PPO and ARS \
2.2 Motivating the relevance of research with a paper from 2021 (Dulac-Arnold et al., 2021) is strange given the fast pace of RL research. I would encourage the authors to update the motivation \
2.3 Only on-policy RL algorithms are used in the experiments. It would be interesting to see the performance also with off-policy RL algorithms (mainly SAC). From what I see, this should be mathematically possible. \
2.4 Possible typos: p.2 line 067, "Analytical Solution the Mountain Car" (header section 2); use of in-bracket citations within the text (e.g., p.2 line 077) "problem in (Moore, 1990)"

---

> ### Author Rebuttal · Authors · 2026-03-30
>
> _Thank you very much for your detailed, encouraging feedback and for
> recognizing the contribution of our work to the RL discipline._
>
> ## Questions
>
> **Q1. Why PPO, ARS, REINCORCE as baseline**
>
> The choice of REINFORCE was motivated as being the natural baseline as for
> being _the_ original policy gradient algorithm. ARS was included for two
> reasons: (1) it is the top-performing SOTA RL method on Mountain Car and (2) it
> specifically targets linear policies. In addition, we selected PPO since it is
> one of the most widely used policy gradient methods and a standard baseline in
> contemporary RL benchmarks. For the Aero2 environment, only PPO was
> investigated in prior work. This resulted in two on-policy algorithms and one
> random-search approach.
>
> However, we agree that it would be very interesting to also look at off-policy
> algorithms. Yes, it is indeed mathematically possible to implement the proposed
> Chebyshev approximator model with off-policy RL algorithms like SAC. The
> software architecture of Stable Baselines3 (SB3), however, requires quite some
> effort to integrate new policy classes (e.g., Chebyshev policies) with the
> battery of RL algorithms (PPO, SAC, etc.).
>
> Instead, we adopted our hybrid approach in the most simple fashion by combining
> a Chebyshev representation layer with a single-layer MLP architecture of Stable
> Baselines3. (In case of a 1-dim action space, it is really just one neuron,
> no bias and linear activation.)
>
> We again ran experiments on Mountain Car: To verify consistency, we also re-ran
> PPO and ARS experiments with the hybrid setup, and added SAC. Results (20
> independently trained agents, evaluated over 100 uniform initial states) are
> shown below.
>
> | Algorithm | Mean Reward | Std Reward | Mean Episode Length |
> | --------- | ----------- | ---------- | ------------------- |
> | CH-3 ARS | 98.72 | 0.13 | 402.06 |
> | CH-4 PPO | 98.47 | 0.16 | 354.56 |
> | CH-3 SAC | 97.27 | 0.36 | 352.04 |
>
> We see that both ARS and PPO results align with what we have seen with native
> implementations (cf. Table 4 in submission). CH-SAC follows this line of
> results, still outperforming all MLP SOTA agents (including SAC with MLP).
>
> We thank the reviewer for sparking this discussion. We added these new results,
> which again highlight the benefits of Chebyshev policies, to the appendix.
>
> **Q2. Feasibility bound on $|x_\text{min} - x_*|$.**
>
> Thank you for this hint. The position x_min of the left wall is feasible if and
> only if the car can reach the goal in one stroke at α=1. If the left wall is
> approximately in the interval [-0.651, 0.310] then this problem is infeasible,
> and otherwise the Mountain Car can indeed reach the goal (from any start
> position).
>
> **Q3. Sketching hybrid architecture.**
>
> The multi-variate Chebyshev polynomials as introduced in this submission allows
> us to model maps ℝⁿ → ℝᵐ. Each layer in an MLP is also a function ℝⁿ → ℝᵐ,
> namely a linear map and then a scalar activation function on each output
> dimension. Now we could simply exchange any neural layer in a MLP by
> a Chebyshev layer. If we do this at the first layer, this becomes something
> like a multi-variate Chebyshev feature space. However, for high-dimensional
> tasks, we may first have a neural MLP that reduces the dimension of input space
> (observation/state space in RL), and then have a Chebyshev layer, and then
> maybe a neural MLP again.
>
> In principle, however, there are hardly any limits. We could also think of net
> architectures without fully-connected layers, but Y/forked/branched/Siamese
> architectures, and after each branch we add a Chebyshev layer. This would
> realize a joint representation learning of the input space, after which (closer
> to the output layer, the action space), Chebyshev layers help blowing up
> non-linearity.
>
> ## Comments
>
> **Theory and novel policies.**
>
> It is true that Chebyshev policies could have been invented without solving
> Mountain Car to optimality. However, the unreasonable bad regret of MLP-based
> SOTA was revealed by the analytical analysis. It is still unclear why SOTA RL
> algorithms have difficulties to train MLPs here. This makes the fact that
> Chebyshev policies, which we introduced here, do not show this deficiency, even
> more interesting. In this sense, we find it hard to separate those two
> contributions.
>
> **Range of applications.**
>
> The reviewer rightfully remarked that our approach is focused on
> low-dimensional control tasks. We find the additional experiments we provided
> (see Q3 of MSdb) where we could handle over 2500 polynomial bases quite
> encouraging that we could handle a range of state space dimensionality that is
> highly relevant for real-world control tasks in industry.
>
> Furthermore, we look very much forward on investigating our hybrid approach as
> described above, especially when we apply "Chebyshev layers" in later layers.
>
> **Abbreviations and typos.** Thank you, fixed.
>
> **Updating motivation.** Thanks for this remark. We have added two more surveys
> to strengthen our motivation from 2025 and 2026.

---

> > ### Author Rebuttal · Reviewer_548M · 2026-04-03
> >
> > I thank the authors for the extensive and clear rebuttal addressing all my concerns, despite my high initial score. The additional experiments and the inclusion of SAC make this a very well-rounded paper in my view. I encourage the authors to add the feasibility bound on the mountain car problem to the paper, as its absence may leave readers wondering. In general, I retain my fully positive scores and recommend acceptance of this refreshing work.

---

> > > ### Author Response · Authors · 2026-04-03
> > >
> > > Dear reviewer, thank you very much for your great support and encouragement!
> > >
> > > We very much like to follow your recommendation and add the feasibility bound to our paper! Thank you for helping in improving our submission.

---

### Official Review · Reviewer_FPWh · 2026-03-05

**Soundness:** 4
**Presentation:** 4
**Significance:** 3
**Originality:** 3
**Overall Recommendation:** 5
**Confidence:** 4

**Summary:**

The study derives an analytical solution to the continuous Mountain Car problem, showing standard deep RL agents exhibit large regret and sub-optimal control strategies. Motivated by the simplicity of the analytical optimal policy, the study introduces Chebyshev policies, which substitute MLPs with multi-variate Chebyshev polynomials. Empirical evaluations show that Chebyshev policies achieve reduced regret, require fewer parameters, and improve sim-to-real transfer compared to standard policies in low-dimensional tasks.

**Compliance With Llm Reviewing Policy:**

Affirmed.

**Final Justification:**

This work is clear, solid, and interesting. It introduces novel algorithms and offers new solutions to an existing environment. It should be accepted. I look forward to future work in this line of research.

**Key Questions For Authors:**

1. Although high dimensionality limitation is discussed by the authors, some comparison and examples could be given to show the limit of pure Chebyshev policies. E.g., at what specific state-action dimensionality do you observe MLP (in different sizes) doing better than Chebyshev policies in terms of computational efficiency and training process?
2. The experimental results for the impact of max degree could be shown to give further insights.
3. Question 1 and 2 will give the readers clear insights on: for what problems we can or shouldn’t use Chebyshev policies and how to set the hyperparameters.
4. The study proposes hybrid approaches combining MLPs and Chebyshev layers as a promising future direction to address higher-dimensional problems. I saw this paper (https://arxiv.org/abs/2508.14536v1) adds a Chebyshev layer for featurization or transformation before MLP which is related?
5. Section 2 title missing “to”.

**Limitations:**

The authors have discussed the limitations of their work, specifically about the numerical instability and exponential scaling of high polynomial degrees in multi-variate settings.

**Strengths And Weaknesses:**

**Soundness**: The study is rigorous, math appears solid, and the empirical design robustly tests Chebyshev policies across multiple standard algorithms against fair state-of-the-art baselines.

**Presentation**: The paper is well written and easy to follow and logical. The visualizations of action maps and phase portraits are clear and effective.

**Significance**: The study provides an exact mathematical solution to the continuous Mountain Car problem, closing a long-standing gap. Chebyshev policies provide an alternative policy parameterization for low-dimensional, embedded control systems requiring high sample efficiency and strict real-time constraints.

**Originality:** Solving the exact optimal control for the continuous Mountain Car is a novel. Formulating multi-variate Chebyshev polynomials as differentiable replacements for MLP is somewhat novel in RL.

**Weakness**: A key weakness is the inherent curse of dimensionality in multi-variate polynomials, which restricts the applicability of these policies to low-dimensional environments. Also, it would be nice to have more results for commonly used benchmark problems such as continuous control, mujoco, etc.

---

> ### Author Rebuttal · Authors · 2026-03-30
>
> _Thank you very much for your encouraging review that helped improving our
> submission._
>
> ## Questions
>
> **Q1. Curse of dimensionality.**
> **Q2. Max-degree impact.**
>
> Thank for this question, which motivated us to set up further experiments that
> shed light on this.
>
> For a state space of dimension n and max-degree d, we have (d+1)ⁿ Chebyshev
> polynomials. This rapidly grows with n. For max-degree 3 and a 6-dimensional
> input space leads to 4k polynomials. For the mountain car problem (n=2) and
> d=50, we have 2601 polynomials. (A MLP with layer sizes 2, 64, 64, 1 has 4355
> trainable parameters.)
>
> In the table below we see experimental results with PPO and varying max-degree
> from 1 to 50. We trained for each row 20 agents and evaluated them (over 100
> uniform starting points). We see that after degree 3 the agents perform well,
> and keep the performance quite well also for higher degrees, in fact beating
> all SOTA agents until degree 20.
>
> However, performance (simulation steps/s) of course drops, though not quadratic
> (as the number of polynomials). However, we have not optimized for performance
> yet, e.g., employ dynamic programming strategies on the recursive computation
> scheme of multi-variate Chebyshev polynomials.
>
> Convergence also highly depends on the RL algorithm used. When we switch from
> PPO to classic REINFORCE then already max-degree 10 does not converge anymore.
>
> | Degree | Mean(Agents) | Best_Agent_Mean | Steps/s |
> | :--- | :--- | :--- | :--- |
> | 1 | -0.0975 | -0.0043 | 61.0 |
> | 2 | -0.6159 | -0.3784 | 53.9 |
> | 3 | 88.4433 | **98.9623** | 50.4 |
> | 4 | **97.9296** | 98.1458 | 46.9 |
> | 5 | 97.78 | 98.2048 | 43.0 |
> | 6 | 97.5848 | 98.0292 | 38.5 |
> | … |
> | 19 | 96.6435 | 97.3213 | 16.1 |
> | 20 | 96.6991 | 97.3904 | 15.4 |
> | 30 | 77.2577 | 95.7942 | 9.8 |
> | 40 | 80.4135 | 94.6989 | 6.0 |
> | 50 | 80.7739 | 94.9583 | 4.0 |
>
>
> **Q3. Limits of Chebyshev.**
>
> A limitation of Chebyshev policies can surely be found in large dimensions $n$
> of the state space combined with a high max-degree $d$, which leads to
> $(d+1)^n$ basis polynomials. Although many engineering tasks deal with $n$
> between, say, 1 and 4, more challenging real-world, physical task have $n \ge
> 10$ and well beyond. Then $(d+1)^n$ quickly becomes infeasible.
>
> We would also like to refer to our answer to question Q2 (where MLPs fit
> better) of reviewer SGJD. Quick summary: The uniform convergence of Chebyshev
> polynomials for many problems is an advantage, but can also be a weakness if
> the optimal policy displays very different behavior at different regions of the
> state space. Here MLPs (especially with ReLU activation) can "cut out" regions,
> erect a hierarchical approximation and adaptively localize their
> expressiveness.
>
> In an hybrid approach in future work we would hope to be able to combine both
> advantages.
>
> **Q4. Hybrid approach.**
>
> In the answer to Q3 of reviewer 548M, we sketched our idea for the future
> hybrid approach.
>
> Thank you for the pointer to the arxiv paper. We did not find it at any
> peer-reviewed venue so our response refers to the arxiv version you provided.
>
> There are commonalities to our research, e.g., they also provide evidence that
> MLP approximators are limited for the investigated control tasks. They are also
> interested in Chebyshev polynomials, but interestingly only at _uni-variate_
> polynomials, which they apply to each dimension of the state space in an
> isolated fashion.
>
> As a consequence, they _lose the Stone-Weierstrass approximation theorem_ and
> hence also (at least our) theoretical justification. (The paper does not
> further discuss the theory behind their approach.) As a result, their approach
> effectively boils down to the heuristic idea of adding _some_ non-linear
> feature space transformation. This probably explains the framing of their paper
> title: "Beyond ReLU" activation function. There has been other prior work in
> literature, dating back to the 1980s, where uni-variate Chebyshev functions
> have been investigated as activation functions.
>
> Our multi-variate Chebyshev polynomials form an orthogonal basis of the
> (continuous) policy space, and span a dense subspace by virtue of
> Stone-Weierstraß approximation theorem. For that it is vital to have basis
> elements that _blend all dimensions_ of the state space, which is why we gave
> a multi-variate generalization.
>
> Given that the uni-variate polynomial T₁(x) = x is the identity map, the
> original feature space is contained in the new one, so we expect that adding an
> MLP that worked before does so again. Surprisingly, they also add T_0, which
> would be covered by the bias of the neurons anyways; skipping it would make
> their feature space sparser without losing anything.
>
> What is also surprising: They limit their research to low-dimensional
> _discrete_ state space and _discrete_ action space. The vast majority of
> low-dimensional control problems in real world is continuous. Although they
> focus a lot on Q-learning, it appears to be an unnecessary restriction.
>
> **Q5. Typo.** Thanks!

---

> > ### Author Rebuttal · Reviewer_FPWh · 2026-04-03
> >
> > Thank you for the thorough and detailed response. The provided clarifications have addressed my concerns.

---

> > > ### Author Response · Authors · 2026-04-03
> > >
> > > Thank you very much for the detailed review and the valuable discussion, which helped to improve our work. And thank you for raising the score to accept (5)!

---

### Official Review · Reviewer_SGJD · 2026-03-12

**Soundness:** 3
**Presentation:** 2
**Significance:** 2
**Originality:** 3
**Overall Recommendation:** 5
**Confidence:** 3

**Summary:**

This paper addresses a long-standing problem of the optimal control solution for the Mountain Car problem. The authors analytically derive the optimal solution, which is surprisingly simple. Motivated by this solution, the paper introduces Chebyshev policies, which are universal approximators constructed with multi-variate Chebyshev polynomials, as a replacement of MLP policies for low-dimensional control tasks. Experimental results show that the proposed Chebyshev policies improve the base algorithms (ARS, Reinforce, PPO) on various RL control tasks.

**Compliance With Llm Reviewing Policy:**

Affirmed.

**Final Justification:**

The paper makes two different contributions. While they are loosely related, they are both solid and valuable. I have therefore raised my score to 5.

**Key Questions For Authors:**

1. What specific mathematical properties of Chebyshev polynomials make them well-suited for approximating policies in low-dimensional control tasks like the Mountain Car problem?
2. In what types of control problems or policy approximation scenarios would a standard MLP inherently outperform a Chebyshev policy, even if the dimensionality is low?
3. At what point does the "curse of dimensionality" render Chebyshev policies impractical?
4. On page 4, the authors states that "Theorem 2.4 does not reveal the optimal C", but is it possible to plug the solution \alpha(t) = C \cdot \dot x(t) back into (5) to derive a optimal C?
5. Which activation function did you use in the MLP policies?

**Limitations:**

Based on the computation complexity analysis (Lemma B.1), the Chebyshev policy suffers from the "curse of dimensionality".
It is not clear what characteristics of Chebyshev polynomials make them outperform usual MLP, hence it’s not clear when to adopt this kind of policy.

**Strengths And Weaknesses:**

**Soundness**

The derivation of the analytic solution seems theoretically sound.
The solution turns out to be surprisingly simple, and motivated by this finding, the authors propose Chebyshev policies which are simple approximators but effective.

**Presentation**

The paper mainly consists of two parts, the analytic solution and the Chebyshev policies.

The derivation of analytic solution could be improved by separating the mathematical logic and physical explanations and analogies (like potential and energy).

In the construction of the analytical worst-case policy, the authors introduced some constants whose provenance could be better explained.

From the analytical solution to the Chebyshev policies, it is not obvious that the Chebyshev polynomials are introduced "from first principles", since there are potentially other universal approximators that could have been considered.

**Significance**

The analytic solution of the Mountain Car problem is a neat result, which helps to understand the nature of this control task.

The use of Chebyshev policies as drop-in replacement for MLP policies is effective for the low dimensional control tasks demonstrated in the paper. However, for higher dimensional tasks and for higher polynomial degree this method may be impractical, and for low dimensional tasks it’s not clear that adopting this method is always preferable to MLPs (e.g., in tasks where the optimal policy may not be smooth).

**Originality**

The novel contribution of this paper includes (1) the analytically solution to the Mountain Car problem and (2) the use of multi-variate Chebyshev polynomials to construct policies for low-dimensional control tasks.

The authors may want to discuss Chebyshev polynomials as approximators have already been extensively explored in existing work for various neural networks.

**Other comments**

The paper should be proofread for English (e.g., "facilitate" seems to be used incorrectly, or "need be") and typos (e.g., in-text citations, or "Theorem 2.2).

---

> ### Author Rebuttal · Authors · 2026-03-30
>
> _Thank you very much for your encouraging feedback and for recognizing the
> significance of our work._
>
> ## Questions
>
> **Q1. First principles and suitability of Chebyshev.**
>
> Thank you for this question. First, we do not think that Chebyshev policies are
> in any way special for the Mountain Car Problem. In fact, they converge
> uniformly over the state space.
>
> What makes them beneficial is that they form a dense subset of the space of
> continuous policies. This is effectively a consequence of Stone-Weierstraß for
> real-valued continuous functions over a compact subset of ℝⁿ.
>
> What makes them numerically attractive is that they form an orthogonal basis.
> The best approximation of an optimal policy π* of any RL task (if it would be
> known) is approximated by "projecting" it to these Chebyshev basis polynomials,
> literally through the inner product <π, T> = ∫ π(s) T(s) ds. In this sense, the
> Chebyshev polynomials behave "independent" in the approximation task, leading
> to well-behaved coefficients (and a simple algorithm if π* is really known.)
>
> Next, uni-variate Chebyshev polynomials have the property that their maxima
> within [-1, 1] are attained at 1 and -1. (This leads to minimizing the
> worst-case approximation error.)
>
> **Q2. Where MLP are a better fit.**
>
> The strengths of Chebyshev policies mentioned above, namely power in uniform
> convergence, can also be weakness: Chebyshev policies treat the state space
> equal. In contrast, MLPs (especially with ReLU activation in hidden layers) can
> "cut out" regions and if the MLP has some depth, they can form a "hierarchical"
> decomposition of the state space. This allows MLPs to implement very different
> behavior at different locations, like high variety/frequency policies at one
> part and low variety at other parts.
>
> In many typical real-world control problems, this may not be relevant. However,
> in case of a bang-bang or sliding mode control, or when we expect different
> behaviors in different "modes" then a MLP might be suited much better.
>
> We added this aspect to our discussion of limitations. Thank you for
> re-sparking this aspect.
>
> **Q3. Curse of dimensionality.**
>
> For a state space of dimension n and max-degree d, we have (d+1)ⁿ Chebyshev
> polynomials. To investigate the impact of d, we repeated training of Chebyshev
> policies using PPO on Mountain Car with increasing max-degree from 1 up to 50.
> For each degree, 20 agents were trained independently and evaluated by
> averaging the return over 100 deterministic starting states sampled from the
> initialization range.
>
> Performance improves rapidly once the policy class becomes sufficiently
> expressive and remains stable over a wide range of higher degrees, still
> beating all SOTA agents even with d=20.
>
> | Degree | Mean(Agents) | Std(Agents) | Best_Agent_Mean | Best_Agent_Std |
> | :--- | :--- | :--- | :--- | :--- |
> | 1 | -0.0975 | 0.0842 | -0.0043 | 0.0002 |
> | 2 | -0.6159 | 0.1649 | -0.3784 | 0.2569 |
> | 3 | 88.4433 | 30.117 | **98.9623** | 0.0769 |
> | 4 | **97.9296** | 0.1271 | 98.1458 | 0.2278 |
> | 5 | 97.78 | 0.1565 | 98.2048 | 0.1625 |
> | 6 | 97.5848 | 0.1891 | 98.0292 | 0.1855 |
> ... |
> | 20 | 96.6991 | 0.7013 | 97.3904 | 0.1901 |
> | 30 | 77.2577 | 38.7163 | 95.7942 | 0.331 |
> | 40 | 80.4135 | 19.5194 | 94.6989 | 0.4354 |
> | 50 | 80.7739 | 21.937 | 94.9583 | 0.5967 |
>
> Beginning with degrees around 55, results do not converge any longer. However,
> these results highly depend on the choice of RL algorithm. Classical REINFORCE
> would already fail at max-degree 10.
>
> We have added these experimental results to the appendix of the
> paper. We would like to further refer to our answer of Q3 of reviewer FPWh.
>
> **Q4. Analytical constant $C$ by eq (5).**
>
> The principle idea is right, but the practical obstacle is that for the
> integral in (5) we need to know the turning points ξᵢ, which one obtains
> themselves by forward integrating the optimal solution α(t) = C ⋅ x(t).
>
> We do not have this obstacle for the final stroke from x_min to the goal $x_*$.
> Recalling that $\dot x = \sqrt{-2 U(ξ)}$, we can plug into (5) as suggested,
> but what we get is a formula that involves the integral over U(ξ), which itself
> is not closed-form, but requires forward integrating. We believe that
> a non-numerical solution will be hard to obtain. This is why we resorted to
> determining the constant C numerically.
>
> **Q5. Activation function.**
>
> We utilize the default activation functions of Stable-Baselines3 version 2.3,
> which are Tanh for A2C, ACER, PPO, TRPO, and ReLU for ARS, SAC, DDPG and TD3.
> These SB3 defaults mirror the architectures used by the authors of the original
> algorithms.
>
> ## Comments
>
> **Prior work.**
> We have extended our prior work section on Chebyshev approximators for neural nets.

---

> > ### Author Rebuttal · Reviewer_SGJD · 2026-04-02
> >
> > I thank the authors for their responses, which have adequately addressed my concerns and questions.

---

> > > ### Author Response · Authors · 2026-04-03
> > >
> > > Thank you very much for the valuable feedback.
> > >
> > > May we **add a comment on the significance** of our research: Indeed we have **submitted a patent application** with an industrial company partner in the application **domain of motion control** in industrial automation, where a core of the innovation is based on this submission. We are looking forward to the patent grant in the next, say, 12 to 24 months, based on previous experience.
> > >
> > > If there is anything else we can provide, please let us now. We would really appreciate if you would consider raising your score for our submission.

---

### Official Review · Reviewer_MSdb · 2026-03-13

**Soundness:** 3
**Presentation:** 3
**Significance:** 4
**Originality:** 3
**Overall Recommendation:** 5
**Confidence:** 4

**Summary:**

The authors claim to have derived a closed form control solution for the classic Mountain Car RL problem. They claim that this allows them to define a new general function approximation solution for RL that is denser and more efficient to learn than standard Deep RL approaches.
The new formulation allows for a simple mathematical close-form expression for the policy options.

**Compliance With Llm Reviewing Policy:**

Affirmed.

**Final Justification:**

After reading the author rebuttal to my review I have more more confidence in the soundness of the claims in the paper. The other reviews and discussion have also helped. It is a focussed niche of a problem in RL but it's a solid theoretical result, which could lead to other attempts to do the same on other classic benchmarks, which would be a benefit for the field generally. I have changed my score to Accept.

**Key Questions For Authors:**

### Questions
On page 7 in comparing to optimal control strategies, I found it unclear. Are you comparing to optimal model based control as well, or is this a more general statement about the problem setup.

I find it hard to evaluate the correctness of the claims here, as I am not expert enough in the theory of continuous systems. But I have a concern about whether providing the functional form, in the terms of a Chebyshev polynomial structure is in some way "cheating" compared to a general neural network setup. I'm not even against such cheating, since it certainly can be the case, and has been shown before, that a domain specific, hand coded value function approximation or policy structure, can outperform general neural representations for RL on given domains [1]. The Chebyshev is surely quite general but I wonder if there is some way that it is particularly well tuned for cyclic, continuous control domains such as Mountain Car.

For instance, the tuning of the hyperparameter for different max-degree polynomials is capturing some quite powerful tuning ability here. What would be the equivalently powerful tuning approach for an MLP policy? I feel like merely tuning the number of hidden layers doesn't get at the same power or "compactness"(?) that increasing the policy from degree 5 to degree 6 gets.

[1] Liang, Yitao, et al. "State of the Art Control of Atari Games Using Shallow Reinforcement Learning." _Proceedings of the 2016 International Conference on Autonomous Agents & Multiagent Systems_. 2016.

**Limitations:**

yes

**Strengths And Weaknesses:**

I find it intriguing that the closed form policy would be independent of the start state, demonstrating some kind of ergodic truth in a concrete form. It was a pleasure to read, really interesting ideas and approach for sure.

In this section it would be helpful to expand on what settings cause the policy to escape the safe solution space, that is, are there any general ways to describe what makes a mountain car policy fail?

It would be helpful to compare and discuss more broadly the solution of continuous control problems like mountain car outside the Reinforcement Learning framework. Maybe I missed it, but I feel like there must have been more work on solving this problem completely before, although looking briefly I didn't find quite the same treatment anywhere.

---

> ### Author Rebuttal · Authors · 2026-03-30
>
> _Thank you for recognizing the significance of out work and our theoretical
> insights in general. We hope to spark similar work for other RL benchmark tasks
> in the future._
>
> ## Questions
>
> **Q1.Comparison to optimal control strategies on page 7.**
>
> We are not certain to what comparison the reviewer refers. In case of Table 1,
> π_ana is the analytical optimum as derived in §3. The lines ARS, SAC and PPO
> refer to MLP-based policies (SOTA).
>
>
> **Q2. Are Chebyshev policies general or fine-tuned?**
>
> Chebyshev are universal: From the Stone-Weierstrass theorem it follows that the
> multi-variate Chebyshev polynomials are dense in the space of continuous
> policies over a compact state space. (In fact, we normalize the state space to
> [-1, 1]ⁿ.) Note that this is about uniform convergence (sup norm), i.e., the
> convergence concerns uniformly the entire state space, i.e., it is not tuned to
> a particular part of the state space.
>
> But we also have good convergence rate. The theory behind this are
> generalizations of the Jackson inequality. If we consider a k-times
> continuously differentiable policy π then the approximation error with degree-d
> Chebyshev polynomials is in O(d^(-k)). (The dimension of the state space is
> hidden in the constant.) See Bagby, Bos, Levenberg, Multivariate Simultaneous
> Approximation, Conti Approx. (2002) 18: 569-577.
>
> **Q3. Tuning and impact of max-degree.**
>
> We thank the reviewer for sparking this discussion, which touches a couple of
> important aspects.
>
> When we increase max-degree, the hypothesis space spanned by the Chebyshev
> polynomials becomes larger in a monotonic sense, by adding more orthogonal
> elements. What we expect is that when a certain degree is reached, the
> performance will not further improve as the optimal policy is (roughly)
> contained in this space. We can therefore actually perform a binary search
> for the max-degree that fits well.
>
> For mountain car with PPO it looks like this (see also answer to Q3 of reviewer
> SGJD):
>
> | Degree | Best_Agent_Mean | Best_Agent_Std |
> | :--- | :--- | :--- |
> | 1 | -0.0043 | 0.0002 |
> | 2 | -0.3784 | 0.2569 |
> | 3 | **98.9623** | 0.0769 |
> | 4 | 98.1458 | 0.2278 |
> | 5 | 98.2048 | 0.1625 |
> | 6 | 98.0292 | 0.1855 |
> | 7 | 98.11 | 0.2196 |
> | ... | | |
>
> MLPs with ReLu activation are quite well understood, and they essentially give
> us a continuous, piecewise-linear map ℝⁿ → ℝᵐ. If the landscape of this map
> shall reach a certain complexity, e.g., many pieces of the linear parts,
> complicated piece domains, then the MLP needs breadth and depth. (For instance,
> (Montúfar et al., 2014) showed that the number of these parts grows
> $O(\prod_{i=1}^L n_i / n_0)$, where we have $L$ layers of sizes $n_1, …, n_L$,
> and input dimension $n_0$.)
>
> However, the hyperparameter tuning of $L, n_1, \dots, n_l$ is less straight
> forward than a binary search. On the other hand, what we gain is that MLPs can
> "focus" their expressiveness to certain regions of the state space. (Compare
> our answer to Q2 of reviewer SGJD.)
>
> ## Comments
>
> **Characterizing failing policies.**
>
> Our analysis does provide insight here. We reported on this on page 7, "How do
> the control strategies compare?", but in the interest of available space only
> briefly. More details are given in §C.6.
>
> Note that the state space $\mathcal{S}$ is the phase space of the dynamical
> system in x. A mountain car moves in closed orbits when no action is applied.
> The goal is to create an outward spiral in phase space. This is achieved by
> sign(a) = sign(ẋ). A policy is successful if it follows this condition (for
> a sufficient amount of time). Some agents trained by SOTA agents violate this
> property at times, see ARS in Fig-5 or the bottom row in Fig-11 in the
> appendix.
>
> We can also quantify how far a policy π is from optimality by ‖π - π_ana‖,
> which is an integral over the state space. Parts of the state space are
> irrelevant, so the sup norm would be a bad choice here, the 2-norm makes more
> sense, see Table 1.
>
> We could actually consider the probability distribution of the state occurrence
> given the initial state distribution. This gives a probability measure μ over
> $\mathcal{S}$, which we can use for the integral in ‖π - π_ana‖. This is future
> work, though.
>
> **Prior work on analytical solutions.**
>
> We did extensive literature research and discussed this also within our
> academic network, actually for over a year now. To the best of our knowledge,
> the optimal solution has been unknown. Unfortunately, not knowing the regret
> of policies of modern RL algorithms on popular benchmark problems is indeed the
> common situation. We believe that our scientific community would benefit
> tremendously from insight gained from deeper optimality analysis on other
> benchmark problems, too.

---

> > ### Author Rebuttal · Reviewer_MSdb · 2026-04-02
> >
> > I thank the authors for their details rebuttal. I have increased my score to Accept as explained in my updated review.

---

> > > ### Author Response · Authors · 2026-04-03
> > >
> > > We thank the reviewer very much for the valuable and encouraging review and for increasing the score to accept (5).

---

### Decision · Program_Chairs · 2026-04-30

**Decision:**

Accept (spotlight)

**Comment:**

Reviewers agreed that this paper makes an original contribution to reinforcement learning for low‑dimensional control. The analytical solution to the Mountain Car problem solves a long‑standing problem and provides insight into the optimality gap of modern RL algorithms on a standard benchmark. Building on this analysis, the introduction of Chebyshev policies is a compelling and well‑motivated alternative to neural networks for low‑dimensional control, with clear performance benefits. Experimental results are also convincing.

Although reviewers shared concerns around the scope of applicability of Chebyshev policies given the curse of dimensionality and demanded clarification of when Chebyshev policies are preferable to MLPs, these concerns were addressed in the rebuttal through additional experiments and clearer discussion of limitations.

Overall, the work is technically sound, well written, and likely to be of interest to the RL and control communities. It offers both a useful theoretical result and a practical policy class that others can build upon.